# WRF-Chem v3.9 simulations of the East Asian dust storm in May 2017: modeling sensitivities to dust emission and dry deposition schemes

Yi Zeng[1,2], Minghuai Wang[1,2], Chun Zhao[3], Siyu Chen[4], Zhoukun Liu[1,2], Xin Huang[1,2], Yang Gao[5]

[1]Institude for Climate and Global Change Research and School of Atmospheric Sciences, Nanjing University, 210023 Nanjing, China

[2]Joint International Research Laboratory of Atmospheric and Earth System Sciences & Institute for Climate and Global Change Research, Nanjing University, China

[3] School of Earth and Space Sciences, University of Science and Technology of China, Hefei, 230026, China

[4]Key Laboratory for Semi-Arid Climate Change of the Ministry of Education, College of Atmospheric Sciences, Lanzhou University, Lanzhou, 730000, China

[5]Key Laboratory of Marine Environment and Ecology, Ministry of Education/Institute for Advanced Ocean Study, Ocean University of China, and Qingdao National Laboratory for Marine Science and Technology, Qingdao, 266100, China

*Correspondence to*: Minghuai Wang (minghuai.wang@nju.edu.cn)

## Abstract

Dust aerosol plays an important role in the radiative budget and hydrological cycle, but large uncertainties remain for simulating dust emission and dry deposition processes in models. In this study, we investigated dust simulation sensitivity to two dust emission schemes and three dry deposition schemes for a severe dust storm during May 2017 over East Asia using Weather Research and Forecasting model coupled with chemistry (WRF-Chem). Results showed that simulated dust loading is very sensitive to different dry deposition schemes, with the relative difference of dust loading using different dry deposition schemes range from 20%-116%. Two dust emission schemes are found to produce significantly different spatial distribution of dust loading. The difference of dry deposition velocity in different dry deposition schemes comes from the parameterization of collection efficiency from impaction and rebound effect. An optimal combination of dry deposition scheme and dust emission scheme has been identified to best simulate the dust storm in comparison with observation. The

optimal dry deposition scheme accounts for the rebound effect and its collection efficiency from impaction changes with the land use categories and therefore has a better physical treatment of dry deposition velocity. Our results highlight the importance of dry deposition schemes for dust simulation.

## 1    Introduction

Dust aerosol is an important component in the atmosphere and it can impact many processes of the Earth system. Through
absorbing and scattering shortwave and longwave radiative fluxes, dust can alter the radiative budgets, which is called the direct effect (Chen et al., 2013; Kok et al., 2017; Zhao et al., 2010, 2011, 2012). Acting as cloud condensation nuclei (CCN) and ice nuclei (IN), dust can change cloud properties and precipitation, which is called the indirect effects (Creamean et al., 2013; Demott et al., 2010). Besides, dust aerosol can absorb solar radiation and change the atmospheric stability and therefore cloud formation, which is known as the semi-direct effect (Hansen et al., 1997). Furthermore, natural dust is
important for air quality assessments and has significant impacts on human health (Abuduwaili et al., 2010; Chen et al., 2019; Hofer et al., 2017; Jiménez-Guerrero et al., 2008; Ozer et al., 2007). Although great progress has been made in dust models and dust simulations in recent decades, large uncertainties remain in dust simulations (Huneeus et al., 2011; Prospero et al., 2010; Todd et al., 2008; Uno et al., 2006; Zender et al., 2004; Zhao et al., 2013).

A complete description of dust events includes dust emission, deposition and transport processes. The differences of dust
simulation mainly result from the uncertainties of dust emission, deposition and transport processes in models. One uncertainty is from dry deposition processes. Dry deposition refers to the transport of particles from the atmosphere to the Earth's surface in the absence of precipitation (Seinfeld and Pandis, 2006). In most aerosol modeling, dry deposition velocity $V_d$ is used to calculate the dry deposition flux and $V_d$ is usually modelled using the resistance-based approach (Pryor et al., 2008). In the resistance-based approach, $V_d$ is determined by gravitational settling, aerodynamic resistance and surface
resistance. Surface resistance is determined by collection efficiency from Brownian diffusion, impaction and interception and is corrected for particle rebound. Slinn (1982) proposed a semi-analytical description of particle collection efficiencies based on the wind tunnel studies, and many dry deposition schemes since then are variants of this model (Binkowski and Shankar, 1995; Giorgi, 1986; Peters and Eiden, 1992; Zhang et al., 2001). As the formulations for collection efficiencies

from different dry deposition schemes are derived from measurements that have been obtained under different meteorological conditions and land surface types, there remains a large discrepancy of these formulations between different dry deposition models (Petroff et al., 2008).

At present, the comparisons of different dry deposition schemes with reliable field measurements are mainly focused on one-dimensional dry deposition models (Hicks et al., 2016; Khan and Perlinger, 2017; Petroff et al., 2008; Ruijrok et al., 1995). For example, Hicks et al. (2016) compared five deposition models with observations and found that $V_d$ predicted for particles less than 0.2 μm is consistent with the measurements, but predicted $V_d$ can vary greatly in the size range of 0.3 to about 5 μm. However, few studies have been conducted to study how different dry deposition schemes affect aerosol concentrations and their spatial distribution in the 3D numerical models. Wu et al. (2018) compared the effects of different dry deposition schemes on black carbon simulation in a global climate model (CESM-CAM5), but did not examine how different dry deposition schemes affect aerosol concentrations for large-size aerosol particles (e.g., diameters > 2.5 μm), such as dust.

Another uncertainty of dust simulation is the treatment of dust emission process in models. Natural dust is typically emitted from dry, erodible surfaces when the wind speed is high. Dust emission process is closely related to soil texture, soil moisture content, surface conditions, atmospheric stability and the wind velocity (Marticorena and Bergametti, 1995). Dust emission schemes are used to predict the dust emission flux and to describe the dust size distribution. Many studies have compared and evaluated the performance of different dust emission schemes (Kang et al., 2011; LeGrand et al., 2019; Su and Fung, 2015; Wu and Lin, 2013, 2014; Yuan et al., 2019; Zhao et al., 2010, 2006). These studies show large diversity of simulated dust emission flux among different dust emission schemes. Zhao et al. (2006) implemented two dust emission schemes in the NARCM (Northern Aerosol Regional Climate Model) regional model, and found that both schemes captured the dust mobilization episodes and produced the similar spatial distributions of dust loading over East Asia, but significant differences exist in the dust emission fluxes and surface concentrations. Kang et al. (2011) compared three dust emission schemes in WRF-Chem and found that the difference between the vertical dust fluxes derived from the three emission schemes can reach to several orders of magnitude. Yuan et al. (2019) found that one scheme strongly underestimated the dust emission while another two schemes can better show the spatial and temporal variation of dust AOD based on

WRF-Chem simulation of a storm outbreak in Central Asia. In another WRF-Chem study, Chen et al. (2017a) concluded that

the dust emission differences mainly come from the dust emission flux parameterizations and differences in soil and surface

input parameters in different dust emission schemes.

While dust emission schemes have been studied quite extensively, few studies have examined dust emission and dry

deposition schemes simultaneously. As both dust emission schemes and dry deposition schemes contribute significantly to

the uncertainties in dust simulations, evaluating dust emission schemes based on a single dry deposition scheme may be

problematic, especially if the dry deposition schemes employed have deficiency. For example, as a widely used regional

model that has been coupled with a variety of dust emission schemes, the WRF-Chem model has been used in many studies

to evaluate the performance of dust emission schemes (LeGrand et al., 2019; Su and Fung, 2015; Wu and Lin, 2013, 2014;

Yuan et al., 2019). But most of these studies use the GOCART aerosol scheme and only one dry deposition scheme (Wesely

et al., 1985) is coupled within the GOCART aerosol scheme. Zhang et al. (2019) compared the modelled dust deposition

using the GOCART aerosol scheme in WRF-Chem with observed dust deposition, and found that modelled dust deposition

is highly underestimated by more than one order of magnitude compared to the observed deposition. This indicates that the

dry deposition scheme (Wesely et al., 1985) in GOCART aerosol scheme may not be suitable for dust simulation and needs

to be further improved.

In this study, we adopted the MOSAIC aerosol scheme coupled within the WRF-Chem model to study how dry deposition

schemes and dust emission schemes affect dust simulations by evaluating model results against observations. As the

MOSAIC aerosol scheme includes several different dry deposition schemes, this allows us to choose more advanced dry

deposition schemes. As the default MOSAIC aerosol scheme only includes the GOCART dust emission scheme, we further

implemented the dust emission scheme Shao2011 (Shao et al., 2011) in the MOSAIC aerosol scheme, which allows us to

compare these two widely used dust schemes along with multiple dry deposition schemes. The goals of this study are: (1) to

study dust simulation sensitivity to different dust emission schemes and dry deposition schemes, (2) to explore which

combination of dust emission scheme and dry deposition scheme can better simulate dust storms in East Asia. The paper is

organized as follows. Sect. 2 introduces the WRF-Chem model, dust emission schemes and dry deposition schemes used,

experiments design and measurements. Sect. 3 analyzes the dust simulation sensitivity to dust emission schemes and dry

deposition schemes and the comparisons with observations. Sect. 4 is the summary and discussion.

## 2    Methodology and measurements

### 2.1    Model description

In this study, WRF-Chem version 3.9 is used. WRF-Chem is built based on the regional mesoscale model WRF, and fully coupled with gas and aerosol chemistry module (Grell et al., 2005). A summary of the settings used to configure the model is listed in Table 1. The Noah land surface model (Chen and Dudhia, 2001) and the Yonsei University (YSU) planetary boundary scheme (Hong et al., 2006) are used in this study. The global soil categorization data set from the United States Geological Survey (USGS) with 24 land categories are used. The Rapid Radiative Transfer Model for General Circulation (RRTMG) radiation scheme (Iacono et al., 2008) is used to calculate the longwave and shortwave radiation. The Grell-Freitas convective scheme (Grell and Freitas, 2014) and the Morrison two-moment microphysics scheme (Morrison et al., 2008) are used. The gas-phase chemistry module used is the Carbon-Bond Mechanism version Z (CBMZ, Zaveri and Peters, 1999). The aerosol module used here is the Model for Simulating Aerosol Interactions and Chemistry with 4 bins (MOSAIC 4-bin) (Zaveri et al., 2008). The MOSAIC 4-bin aerosol scheme divides airborne particles into four size bins by their effective diameter (0.039-0.156, 0.156-0.625, 0.625-2.5, 2.5-10.0 μm) to represent aerosol size distribution. The first three bins represent the Aitken mode and accumulation mode of aerosol. The last bin represents the coarse mode of aerosol. The MOSAIC aerosol scheme includes sulfate, methane sulfonate, nitrate, chloride, carbonate, ammonium, sodium, calcium, black carbon (BC), primary organic mass (OC), liquid water and other inorganic mass (OIN). The OIN species include silica, other inert minerals and trace metals. The emitted dust is assigned to the OIN class of MOSAIC to simulate the major aerosol processes. To study the sensitivity of dust simulation to different dust emission schemes and dry deposition schemes, we test two different dust emission schemes (see Sect. 2.2) and three dry deposition schemes (see Sect. 2.3) within MOSAIC.

### 2.2    Dust emission schemes

Dust emission schemes include empirical schemes and schemes based on dust physical processes. Because of differences in input parameters and formulas to calculate dust flux, dust emission varies among different dust emission schemes. The

Goddard Chemistry Aerosol Radiation and Transport (GOCART) dust emission scheme (Ginoux et al., 2001) is an empirical scheme and was implemented in MOSAIC by Zhao et al. (2010). The GOCART dust emission scheme within MOSIAC aerosol scheme is called by setting dust_opt=13. The University of Cologne (UoC) dust emission schemes (Shao, 2001, 2004; Shao et al., 2011) (Shao schemes) are size-resolved dust emission scheme based on the wind erosion physical theory. The UoC dust emission scheme within GOCART aerosol scheme is called by setting dust_opt=4. When the UoC dust emission scheme is selected, the user should also choose one of the UoC sub-options by setting dust_scheme=1 for Shao2001, dust_schme=2 for Shao2004, or dust_schme=3 for Shao2011. The Shao dust emission schemes are widely used for dust simulations in East Asia, and have been found to perform well in simulating dust emission fluxes (Shao et al., 2011; Su and Fung, 2015; Wu and Lin, 2014), and Shao2011 (Shao et al., 2011) is a simplified version of Shao2004 (Shao, 2004). To test the sensitivity of dust simulation to different dust emission schemes, we implemented the Shao2011 dust emission scheme in MOSAIC aerosol scheme. Each dust emission scheme is described in detail below.

### 2.2.1 GOCART

The formula of vertical dust flux in GOCART is approximated as:

$$F_p = \begin{cases} CSs_p u_{10}^2 (u_{10} - u_t) & \text{if } u_{10} > u_t \\ 0 & \text{otherwise} \end{cases} \tag{1}$$

where C is an empirical proportionality constant, and S is the source function that is determined by the erodibility factor (see Fig.1). $s_p$ is the fraction of each size class of the emitted dust. $u_{10}$ is the horizontal wind speed at 10 meters. $u_t$ is the threshold velocity below which the dust emission does not occur. $u_t$ is calculated as:

$$u_t = u_{t0} * (1 + 1.2 \, log_{10} \, w) \tag{2}$$

where $u_{t0}$ is the threshold velocity for dry soil and w is the soil surface wetness. The formula of $u_{t0}$ is not from the original GOCART paper (Ginoux et al., 2001), but from Marticorena and Bergametti (1995) :

$$u_{t0} = 0.129 \frac{\left(\frac{\rho_p g d_p}{\rho_a}\right)^{0.5} \left(1 + \frac{0.006}{\rho_p g d_p^{2.5}}\right)^{0.5}}{\left[1.928 \left(a(d_p)^x + b\right)^{0.092} - 1\right]^{0.5}} \tag{3}$$

where $\rho_p$ is the density of particles, $\rho_a$ is the density of air, $d_p$ is particle diameter, a equals 1331, x equals 1.56 and b

equals 0.38. The original GOCART dust emission scheme in GOCART aerosol scheme (dust_opt=1) calculates the dust emission flux from 0.2 to 20 μm. For GOCART dust scheme in MOSAIC aerosol scheme (dust_opt=13), the total dust emissions from 0.2 to 20 μm are redistributed to the size bins of MOSAIC (0.039-0.156, 0.156-0.625, 0.625-2.5 and 2.5-10.0 μm) with mass fractions of 0%, 0.38%, 8.8%, 68.0% (Kok, 2011; Zhao et al., 2013). We note that in addition to the size distribution, the values of empirical proportionality constant C are also different for the two GOCART dust emission scheme options. For dust_opt=13, C value is set to $1.0 \times 10^{-9}$ kg s$^2$ m$^{-5}$, which is consistent with the original GOCART dust emission scheme paper (Ginoux et al., 2001). For dust_opt=1, C value is set to $0.8 \times 10^{-9}$ kg s$^2$ m$^{-5}$.

### 2.2.2   Shao2011

The Shao2011 dust emission scheme is a size-resolved dust emission scheme based on the wind erosion physical theory. The dust flux is determined by:

$$F(d_i) = c_y \eta_{mi} (1 + \sigma_m) \frac{gQ}{u_*^2} \tag{4}$$

where $F(d_i)$ is the dust emission rate of particle size $d_i$; $c_y$ is the dimensionless coefficient; $\eta_{mi}$ is the mass fraction of free dust for a unit soil mass; $\sigma_m$ is bombardment efficiency; $Q$ is the saltation flux averaged over the range of sand particle sizes. In Shao2011, the erodibility factor is only used to constrain the potential emission regions. Dust emission is permitted in Shao2011 where the erodibility factor is greater than zero. As the Shao2011 scheme is a size-resolved dust emission scheme, it first calculates the emitted dust from 0.98 um to 20 μm with 40 size bins. Dust emissions from these 40 size bins are then grouped into the four size bins of the MOSACI aerosol scheme (0.039-0.156, 0.156-0.625, 0.625-2.5, 2.5-10 μm). The details of the Shao2011 dust emission scheme are described in Appendix A. There is a bug in calculating dust emission flux in Shao2011 scheme reported after WRF-Chem v3.9, and we have already corrected it in our simulation (See Appendix A). We should mention that the Shao2011 dust emission scheme used in this study is based on WRF-Chem v3.9 with some modifications from WRF-Chem v3.7.1. The difference of Shao2011 among different WRF-Chem versions are documented in Appendix B.

### 2.3   Dry deposition schemes

For dry deposition schemes, dry deposition velocity ($V_d$) is used to calculate dry deposition flux. $V_d$ is determined by gravitational settling velocity ($V_g$), aerodynamic resistance ($R_a$) and surface resistance ($R_S$). There are three dry deposition schemes available in WRF-Chem coupled with the MOSAIC module and used in this study as referred to BS95 (Binkowski and Shankar, 1995), PE92 (Peters and Eiden, 1992) and Z01 (Zhang et al., 2001). Each dry deposition scheme will be described in detail below.

### 2.3.1 BS95

In the BS95 scheme (Binkowski and Shankar, 1995), $V_d$ is expressed as:

$$V_d = V_g + \frac{1}{R_a + R_s + R_a R_S V_g} \tag{5}$$

where $R_a$ and $R_s$ are aerodynamic and surface resistance; $V_g$ is the gravitational settling velocity and is given as:

$$V_g = \frac{\rho_p d_p^2 g C_c}{18\mu} \tag{6}$$

where $C_c$ is the Cunningham correction factor as a function of $d_p$ and mean free path of air ($\lambda$), and $\mu$ is the viscosity dynamic of air. The surface resistance is calculated as:

$$R_s = \frac{1}{u_*(E_B + E_{IM})} \tag{7}$$

where $E_B$ is collection efficiency from Brownian diffusion. $E_B$ is calculated as follows:

$$E_B = Sc^{-\frac{2}{3}} \tag{8}$$

where Sc is the Schmidt number, given by $Sc = \nu/D$. $\nu$ is the kinematic viscosity of air and D is the particle Brownian diffusivity. $E_{IM}$ is the collection efficiency due to impaction of the particle with the collecting surface (Gallagher, 2002). Impaction occurs when there are changes in the direction of airflow, and particles that cannot follow the flow will collide with the obstacle and stay on the surface due to the inertia (Giardina and Buffa, 2018). $E_{IM}$ is given by:

$$E_{IM} = 10^{-\frac{3}{St}} \tag{9}$$

where St is the Stokes number, given by:

$$St = \frac{u_*^2 v_g}{g\nu} \tag{10}$$

St is the ratio of the particle stop distance to the characteristic length of the flow and describes the ability of particles to adopt the fluid velocity (Pryor et al., 2008; Seinfeld and Pandis, 2006).

### 2.3.2   PE92

In PE92 scheme (Peters and Eiden, 1992), the dry deposition velocity ($V_d$) is expressed as:

$$V_d = V_g + \frac{1}{R_a + R_S} \tag{11}$$

The formula of $V_g$ and $R_a$ is the same as in BS95, but the way to calculate $R_S$ is quite different. In PE 92, $R_S$ is parametrized as:

$$R_S = \frac{1}{u_*(E_B + E_{IM} + E_{IN})R} \tag{12}$$

where $E_{IN}$ is collection efficiency from interception and R is the factor for particle rebound. $E_{IM}$, $E_{IN}$ and R are expressed as:

$$E_{IM} = \left(\frac{St}{0.8 + St}\right)^2 \tag{13}$$

$$E_{IN} = \frac{(0.0016 + 0.0061z_0)d_p}{1.414 \times 10^{-7}} \tag{14}$$

$$R = e^{-2\sqrt{St}} \tag{15}$$

$z_0$ is the roughness length and $d_p$ is particle diameter. Stokes number is given by:

$$St = \frac{\rho_p d_p^2}{9\mu d_c} u \tag{16}$$

u is the horizontal wind velocity, $d_c$ is the diameter of the obstacle.

### 2.3.3   Z01

In Z01 scheme (Zhang et al., 2001), the formula of $V_d$ is the same as in BS95 scheme (Eq. (5)). Surface resistance $R_S$ is calculated as:

$$R_S = \frac{1}{\epsilon_0 u_*(E_B + E_{IM} + E_{IN})R} \tag{17}$$

$$E_B = Sc^{-\gamma} \tag{18}$$

where $\gamma$ depends on land use categories (LUC) and lies between 0.50 and 0.58.

$E_{IM}$ is expressed as:

$$E_{IM} = \left(\frac{St}{\alpha+St}\right)^{\beta} \tag{19}$$

where $\beta$ equals to 2. $\alpha$ depends on LUC and lies between 0.6 and 100.0. The Stokes number is given by:

$$St = V_g u_*/gA \tag{20}$$

over vegetated surfaces (Slinn, 1982) and

$$St = V_g u_*^2/g\nu \tag{21}$$

over smooth surfaces or surfaces with bluff roughness elements (Giorgi, 1988). $E_{IN}$ is the collection efficiency based on the relative dimensions of the particle to the collector diameter (Gallagher, 2002). Interception occurs when particles moving with the mean flow and the distance between an obstacle and particle center is less than half of the diameter. Then the particles will collide with and be collected by the obstacle. $E_{IN}$ is expressed as:

$$E_{IN} = \frac{1}{2}\left(\frac{d_p}{A}\right)^2 \tag{22}$$

over vegetated surfaces and $E_{IN} = 0$ for non-vegetated surfaces, where A is the characteristic radius of collectors. A depends on LUC and lies between 2.0 mm and 10.0 mm. R is expressed as:

$$R = e^{-1.0\sqrt{St}} \tag{23}$$

The main difference of formulas used to calculate dry deposition velocity for three different dry deposition schemes are listed in Table 2. For $R_S$, PE92 and Z01 include the collection efficiency from interception ($E_{IN}$) and the rebound effect (R), while these two are neglected in BS95. For the $E_{IM}$ parameterization, all three schemes use St to parameterize $E_{IM}$, but the formulas are quite different. BS95 has a different formula from PE92 and Z01, while the PE92 and Z01 have the same formula but with different coefficients. For PE92, the coefficient for $E_{IM}$ is constant for all the surface types. For Z01, the coefficients $\alpha$ and $\beta$ for $E_{IM}$ change with different surface types. For the $E_{IN}$ parameterization, BS95 ignores this effect; PE92 and Z01 use different formulas and variables to calculate $E_{IN}$. When large particles (usually >5 μm) hit the non-sticky surface, they are liable to rebound from the surface if they have sufficient kinetic energy. The rebound factor R represents

the fraction of particles that stick to the surface (Seinfeld and Pandis, 2006). For rebound effect, BS95 does not consider it; PE92 and Z01 use the same e-exponential form $e^{-b\sqrt{St}}$ to calculate the rebound effect with different coefficient b. For PE92, b is 2.0; for Z01, b is 1.0. In addition, the parameterization of St is quite different for different dry deposition schemes. For BS95, the formulation of St tends to emphasize the nature of the flow field (Binkowski and Shankar, 1995; Pryor et al., 2008). For Z01, the formulation of St is from Slinn (1982) over vegetated surfaces and from Binkowski and Shankar (1995) over smooth surfaces. The formulation of St from Slinn (1982) and Peters and Eiden (1992) are focus on the individual obstacles (Pryor et al., 2008).

## 2.4 Experiments Design

We use WRF-Chem v3.9 with 20 km × 20 km horizontal resolution and 35 vertical levels with model top pressure at 50hPa. The domain covers most of the East Asia (14-60°N, 74-130°E) as shown in Fig. 1. The simulation period is from 26 April to 7 May 2017 with time step of 60s and frequency of output every hour. The timestep between radiation physics calls is 20 minutes. During this period, a severe dust storm event originated from northwestern China and Outer Mongolia, and air quality deteriorated dramatically in a very short time in downwind areas (Guo et al., 2019; Zhang et al., 2018). Meteorological conditions are initialized and forced at the lateral boundaries using the 6-hourly National Center for Environmental Prediction Final (NCEP/FNL) Operational Global Analysis data at a resolution of 1°×1°. For meteorological conditions (such as wind speed and temperature), we reinitialized every 24 hours using NCEP/FNL reanalysis data. For chemistry, the output of the aerosol field (such as the concentration of different aerosol species) from the previous 1-day run was used as the initial chemical conditions for the next 1-day run. Our simulation period is from 26 April to 7 May 2017 and one experiment consists of 12 one-day runs. In this way, the chemical field are continuous and we can also get more reliable meteorological conditions. The MOSAIC aerosol scheme was used for all the simulations. Simulation results prior to 28 April are treated as model spin up for chemical initial condition and are not included in results presented in Sect. 3. The model results from 1 May to 7 May are used for the dust loading and concentration analysis. And the model results from 28 April to 7 May are used for the dust emission analysis as the dust emission before 1 May also have influence on the dust concentration during 1 May to 7 May. To study the dust simulation sensitivity to dust emission and dry deposition schemes,

we run 6 experiments with two different dust emission schemes and three dry deposition schemes (See Table 3). The corresponding model configuration for dry deposition processes of the six experiments also listed in Table 3. We note here that the USGS LUC should be selected for Z01 dry deposition scheme.

## 2.5 Measurements

### 2.5.1 PM10

Hourly surface observed PM10 is used to compare with the simulated PM10 from WRF-Chem. In China, hourly surface PM10 concentrations were collected from more than 1000 environmental monitoring stations (locations shown in results section) maintained by the Ministry of Environmental Protection (MEP). The hourly PM10 data from 1 May to 7 May, 2017

were downloaded from http://beijingair.sinaapp.com/. We collocated the PM10 data to WRF-Chem simulation grids to evaluate model performance with different configurations.

### 2.5.2 MODIS AOD

Daily aerosol optical depth (AOD) from Moderate Resolution Imaging Spectroradiometer (MODIS) is used to compare with our simulated AOD from WRF-Chem. The MODIS onboard Aqua satellite was launched by the NASA in 2002 and Aqua is a

part of A-Train satellite constellation. To compare modelled AOD with observations, we use AOD retrievals at 550 nm from MODIS AOD products on Aqua with daily gridded data at a resolution of 1°×1° (MYD08_D3, Collection 6, combined dark target and deep blue AOD). The MODIS Aqua collection daily MYD08_D3 files were obtained from https://ladsweb.nascom.nasa.gov. As Aqua passes through every region of Earth at around 13:30 local time, we extract the model simulation results at 13:00 to compare with the daily MODIS AOD. For the model results, first we divided the domain

into different time zones according to the longitude. Then the model results at corresponding UTC when the local time is 13:00 are extracted. The collocated model AOD results for each day are used to compared with daily MODIS AOD.

### 2.5.3 CALIPSO data

The vertical profile of aerosol extinction coefficient at wavelength of 532 nm from the Cloud-Aerosol Lidar and Infrared

Pathfinder Satellite observation (CALIPSO) satellite is used to evaluate model results. The CALIPSO launched on 28 April

2006 equipped with CALIOP (Cloud-Aerosol Lidar with Orthogonal Polarization). The CALIOP lidar provides an along-track

observation of aerosol and cloud vertical profile. The vertical and horizontal resolutions for the CALIOP from the surface to

8.2 km are 30 m and 333 m, respectively. Above 8.2 km, the vertical and horizontal resolutions are 60 m and 1 km, respectively.

We use the CALIPSO level 2 APro product (V4.20) to obtain the aerosol extinction coefficient

(CAL_LID_L2_05kmAPro-Standard-V4-20). The CALIPSO data are available at: https://www-calipso.larc.nasa.gov/.

**3    Results**

**3.1    Meteorological conditions**

Dust emission and transport processes are closely related to the meteorological conditions. So we first evaluated the model

performance in simulating the synoptic conditions. Figure 2 shows the surface meteorological conditions during the dust event.

Panels (a)(d)(g)(j) of Fig.2 show the daily mean wind field at 10 meters and daily mean temperature at 2 meters from

NCEP/FNL reanalysis data. The meteorological conditions at 700 hPa show in the supplementary materials (Fig. S1). This dust

storm was triggered by the development of a Mongolian cyclone (Fig. S1c and Fig.2d). With the strong northwest and

southwest wind near the dust source region, emitted dust was transported to the southeast and northeast of China (Fig. S1c, S1e

and Fig. 2d, 2g). Panels (b)(e)(h)(k) of Fig. 2 show the WRF-Chem simulated daily mean wind field at 10 meters and daily

mean temperature field at 2 meters. Panels (c)(f)(i)(l) of Fig.2 show the difference of daily mean wind speed at 10 meters

between WRF-Chem simulation and NCEP/FNL reanalysis data. The WRF-Chem model was able to simulate the wind speed

well over the dust source regions (the Taklimakan Desert and the Gobi Desert) and the eastern and southern China, where the

differences were mostly in the range of -2.0 – 2.0 m s$^{-1}$. The wind speed is slightly underestimated near the center of the

cyclone (Fig. 2c, 2f, 2i, 2l) and as this is away from dust source regions, we do not expect this underestimation causes large bias

in dust emissions. The correlation coefficient (R) and root mean square error (RMSE) between WRF-Chem simulation and

FNL reanalysis data for temperature at 2 meters, U component of wind, V component of wind and wind speed at 10 meters

during simulation period are shown in Table 4. The R for time-averaged temperature at 2 meters, U component of wind, V

component of wind and wind speed at 10 meters from 1 May to 7 May are 1.0, 0.90, 0.86 and 0.82, respectively. The RMSE for time-averaged temperature at 2 meters, U component of wind, V component of wind and wind speed at 10 meters from 1 May to 7 May are 1.03, 1.08, 0.98 and 1.11, respectively. The R for temperature, U component of wind, V component of wind and wind speed at 700 hPa from 1 May to 7 May are 1.0, 0.94, 0.91 and 0.95, respectively (Table S1). The RMSE for temperature, U component of wind, V component of wind and wind speed at 700 hPa from 1 May to 7 May are 0.67, 2.34, 2.70 and 1.76, respectively (Table S1). Overall, the correlation coefficients are generally large and the RMSEs are generally small. This indicates that the WRF-Chem performed well in simulating the meteorological conditions. We also compared the difference of the meteorological conditions in our six experiments and found that the difference is negligible (Fig. S2 and Table S2).

## 3.2    Dust simulation sensitivity to dust emission schemes

In this section, we examine the changes of the simulated dust loading using different dust emission schemes. Figure 3 shows simulated mean dust loading for six experiments over the 7-day simulation period 1-7 May, 2017. When using the same dry deposition scheme (BS95, PE92 or Z01), different dust emission schemes give very different dust spatial distribution. Compared with the Shao2011 scheme, GOCART has higher dust loading over the Taklimakan desert (TD) but has relatively lower dust loading over the Gobi Desert (GD), the south of Outer Mongolia and most parts of northern China. The difference of the spatial distribution of dust loading is mainly caused by the different spatial distribution of dust emission flux from dust emission schemes, as shown in Fig. 4. As the dust emission before 1 May also have influence on the dust loading during 1 May to 7 May, the total dust emission from 28 April to 7 May are analyzed. The total dust emission from 00:00 UTC 28 April to 23:00 UTC 7 May over the GD from GOCART and Shao2011 are 4.90 Tg and 13.88 Tg, respectively. The total dust emission from 00:00 UTC 28 April to 23:00 UTC 7 May over the TD from GOCART and Shao2011 are 7.16 Tg and 2.75 Tg respectively. Over the GD, Shao2011 scheme has higher dust emission than GOCART; while over the TD, GOCART scheme has higher dust emission than Shao2011 (Fig. 4c).

The first column of Fig. 5 shows the spatial distribution of friction velocity, threshold friction velocity, the difference between friction velocity and threshold friction velocity and the dust emission flux from Shao2011 at 06:00 UTC on 3 May. The areas where the friction velocity is greater than the threshold friction velocity is mainly located in the west inner

Mongolia and the south of Outer Mongolia (Fig. 5e). This is consistent with Fig. 5g. When the friction velocity is larger than threshold friction velocity, dust can be emitted from the surface. The second column of Fig. 5 shows the spatial distribution of wind speed at 10 meters, threshold velocity, the difference between wind speed at 10 meters and threshold velocity and the dust emission flux from the GOCART dust emission scheme. Different from Shao2011, the dust emission regions from

GOCART are not only determined by wind speed, but also constrained by erodibility factor (Eq. (1)). From Fig. 5f, the threshold velocity is much smaller than the wind speed at 10 meters in most areas. In these areas, GOCART use Eq. (1) to calculate the dust emission flux, and the source function S depends on the erodibility factor. The dust emission flux in GOCART is directly scaled by erodibility factor. Figure 1 shows the erodibility factor which describes the fraction of erodible surface in each grid cell. As shown in Fig. 5h, dust emission occurs where the wind speed is high and the erodibility

factor is larger than 0.

Over the TD, Shao2011 produces lower dust emission flux than GOCART. One reason may be the formula used to calculate the threshold velocity (Eq. (3)). The formula used to calculate threshold velocity is from Marticorena and Bergametti (1995), which was originally designed to calculate threshold friction velocity (see LeGrand et al. (2019) for details). This inconsistency leads to very small threshold velocity in GOCART, which may result in dust emission at low wind speed.

Another reason may be the incorrect soil particle size distribution over the TD (Wu and Lin, 2014). The incorrect soil particle size distribution can lead to the unreasonable dust emission flux in Shao2011 over the TD. Over the GD, the GOCART scheme has lower dust emission than the Shao2011 scheme. As mentioned by Su and Fung (2015), the erodibility factor over the GD is highly underestimated and need to be improved for the GOCART dust emission scheme.

As we mentioned in Sect. 2.2.2, the Shao2011 used in this study is based on WRF-Chem v3.9 with some modifications from

WRF-Chem v3.7.1. The modified Shao2011 simulates better dust loading than the original Shao2011 scheme in WRF-Chem v3.9 (not shown). Simulated dust emission fluxes are quite different when using two versions of the Shao2011 scheme in WRF-Chem v3.9 and WRF-Chem v3.7.1, which is mainly caused by different soil particle size distributions in two versions. The differences of Shao2011 among different WRF-Chem versions are documented in Appendix B.

### 3.3  Dust simulation sensitivity to dry deposition schemes

In this section, we analyze dust simulation sensitivity to different dry deposition schemes using the six experiments. For simulated dust loading using the GOCART dust emission scheme (the first row in Fig. 3), compared to the BS95 dry deposition scheme, PE92 and Z01 produce higher dust loading over the dust source regions and remote regions. The relative difference of mean dust loading from PE92 and Z01 relative to BS95 is 20% and 59% respectively. As for the simulated dust loading using the Shao2011 dust emission scheme (the second row in Fig. 3), PE92 and Z01 schemes also produce higher

dust loading than BS95 scheme, and the relative difference to BS95 is 72% and 116% respectively. This indicates that dust simulation is very sensitive to dry deposition schemes.

Figure 6a shows the modeled dry deposition velocity over desert surface. As desert dust mass is mainly concentrated in the large particle size range, our dry deposition analysis focuses on the coarse mode (2.5-10 μm) (Kok, 2011; Zhao et al., 2013). The reference diameter of the coarse mode is defined at 5 μm (Fig. 6). BS95 produces larger $V_d$ than PE92 and Z01 in the

coarse aerosol mode. Larger $V_d$ leads to larger dry deposition and thus lower dust loading, consistent with the lower simulated dust loading from the BS95 scheme discussed above (Fig. 3). In Eq. (5), the dry deposition velocity is comprised of gravitational velocity, aerodynamic resistance and surface resistance. The diversity of different dry deposition schemes mainly comes from the way to parameterize surface resistance, and differences from gravitational settling and aerodynamics resistance are small (not shown), consistent with previous studies (e.g., Bergametti et al., 2018). Figure 6b shows the surface

resistance from different schemes as a function of particle diameter ($d_p$). In the coarse aerosol mode, Z01 produces the largest surface resistance, followed by PE92 and BS95. Larger surface resistance causes smaller dry deposition velocity in Z01, thus larger dust concentration as shown in Fig. 3.

The surface collection efficiency is comprised of Brownian diffusion, impaction, and interception and is corrected for particle rebound (see Eq. (12)). Collection from Brownian diffusion is most important for the smaller particles while

collection from impaction and interception play a more important role for large particles in surface collection processes. Figure 6c shows the surface collection efficiency from impaction ($E_{IM}$) from different schemes as a function of particle diameter. BS95 gives the largest $E_{IM}$ and Z01 gives the smallest. Based on field observation data, Slinn (1982) used a semi-empirical fit for smooth surface (Eq. (9)), and Binkowski and Shankar (1995) adopted this formula for $E_{IM}$ and used it for all land surface types. Peters and Eiden (1992) uses Eq. (19) to describe $E_{IM}$, with $\alpha$ equals to 0.8 and $\beta$ equals to 2 to

get the best fit for the data collected over a spruce forest (Eq. (13)). In Zhang et al. (2001) scheme, $\alpha$ varies with LUC and

$\beta$ is chosen as 2 (Eq. (19)). For BS95 and PE92, the formula of $E_{IM}$ is derived from a specific land surface type, but they

have been applied to all land surface types in WRF-Chem. This may lead to large uncertainties for dry deposition over the

whole domain with different surface types. As $E_{IM}$ of Z01 varies with LUC, Z01 may have a better physical treatment of

$E_{IM}$ than the other two dry deposition schemes.

Figure 6d shows the surface collection efficiency from interception ($E_{IN}$). $E_{IN}$ depends on the particle diameter and the

characteristic radius of the collectors (Seinfeld and Pandis, 2006). $E_{IN}$ is important for large particles on hairs at the leaf

surface, and is negligible over non-vegetated surface such as the desert surface we analyzed here (Chamberlain, 1967; Slinn,

1982; Zhang et al., 2001). In BS95, the effect of interception is not considered. In the original PE92 scheme as described in

Peters and Eiden (1992), $E_{IN}$ is also not considered. But in the PE92 scheme used in WRF-Chem, $E_{IN}$ increases with

particle diameter as in Eq. (14). In Z01, the effect of interception is considered as Eq. (22) over vegetated surface and is not

considered for non-vegetated surface (as shown in Fig. 6d over desert surface type). The parameterization of $E_{IN}$ partially

results in the difference of surface resistance between PE92 and the other two dry deposition schemes.

Figure 6e shows the rebound factor from different dry deposition schemes. Rebound and resuspension have long been

recognized as a mechanism by which the surface can act as sources of particles (Pryor et al., 2008). Due to limited

knowledge on particle rebound and resuspension processes, most dry deposition models adopted the form of the rebound

effect as $R = e^{-b\sqrt{St}}$ suggested by Slinn (1982) (Zhang and Shao, 2014; Zhang et al., 2001), while some dry deposition

schemes do not include the rebound effect with R=1.0 (Binkowski and Shankar, 1995; Petroff and Zhang, 2010; Zhang and

He, 2014). BS95 does not consider the rebound effect. b is equal to 2.0 for PE92 scheme and 1.0 for Z01 scheme. Another

difference between PE92 and Z01 is the threshold particle diameter for including the rebound effect. Rebound effect is

included for PE92 when particles are larger than 0.625 $\mu m$ and for Z01 when particles are larger than 2.5 $\mu m$. In summary,

the smaller $E_{IM}$ and rebound factor lead to larger $R_S$ in Z01, while the larger $E_{IM}$ leads to smaller $R_S$ in BS95, and the

moderate $E_{IM}$ and rebound effect give a moderate $R_S$ for PE92.

Figure 6f shows the Stokes number from different dry deposition schemes. Over smooth surfaces, the formula of St for BS95

and Z01 is the same, as shown in Eq. (10). In PE92, St is calculated using Eq. (16), which is similar to the formula used in

Slinn (1982). BS95 and Z01 schemes give a larger St than PE92. Stokes number is used to calculate both R and $E_{IM}$. The difference of Stokes number and the different formulas of R and $E_{IM}$ lead to the different R and $E_{IM}$ among different dry deposition schemes (Fig. 6c and 6e).

    Our discussion indicates that Z01 has a better physical treatment of dry deposition velocity, as Z01 considers the rebound effect and $E_{IM}$ changes with LUC. The Z01 scheme has also been documented to agree better with measured dry deposition

fluxes and dry deposition velocity (e.g., Zhang et al., 2012; Connan et al., 2018). Zhang et al. (2012) compared the dry deposition fluxes measured at five sites in Taiwan with the modeled dry deposition fluxes and found that the measured dry deposition fluxes can be reproduced reasonably well using the Z01 scheme. Connan et al. (2018) conducted experimental campaigns on-site to determine dry deposition velocity of aerosols and found that the Z01 scheme is most suitable for operational use in the size range 0.2-10 μm. All these indicate that the Z01 dry deposition scheme is more physically

meaningful than other two dry deposition schemes.

### 3.4   Comparisons with observations

    To better evaluate the performance of different experiments, we compared the model results with observations. Figure 7 shows hourly observed $PM_{10}$ concentrations over observational sites at 02:00 UTC on 4 May, 2017 (10:00 Beijing Time (BJT) on 4 May, 2017). Very high $PM_{10}$ values ($>1000$ μg m$^{-3}$) are observed in northern China. Figure 8 compares simulated $PM_{10}$

in six experiments with observed $PM_{10}$. During the comparison, the observational sites closest to the model grids are paired up. The correlation coefficients (R), root mean square errors (RMSE) between model and observations, and the mean simulated and observed $PM_{10}$ for all the sites over the five regions during the 7-day period 1-7 May are marked in Fig. 8. The simulated $PM_{10}$ of all the six experiments have obviously underestimated the observations. Among all these experiments, GOBS95 has the lowest average $PM_{10}$ concentration, with a value of 26.45 μg m$^{-3}$, and S11Z01 has the largest one, with a value of 105.17

μg m$^{-3}$, the closest one to the observed mean value of 172.70 μg m$^{-3}$. S11Z01 gives a large R of 0.77 and the smallest RMSE of 96.14 compared to other experiments. Table 5 shows the R and RMSE between the model and observations for $PM_{10}$ for six experiments over five sub regions and over whole China. Over the TD, GOBS95 gives the largest R and smallest RMSE. Over the GD, GOZ01 and S11Z01 gives a better performance compared with other experiments. For other

regions (NCP, NEP and YR), S11Z01 gives a relatively larger R and smallest RMSE. For all the stations in total, S11Z01

gives a larger R of 0.83 and the smallest RMSE of 82.98. Overall, the S11Z01 experiment has the best performance for

simulating this dust storm.

Figure 9 shows the MODIS observed daily mean AOD and WRF-Chem simulated AOD over the simulation period 1-5 May.

For strong dust storms like the one we examined here, dust particles contribute the most to AOD, and AOD therefore can

represent the dust loading in the atmosphere. To match the MODIS AOD observation time, simulated AOD at 13:00 local

time is used for comparison (see Sect. 2.5 for details). For each 1 °×1° grid with observed AOD from MODIS, the average

value of simulated AOD from WRF-Chem in this grid is calculated. Grid points without valid MODIS AOD retrieval are

masked for both observational and model results in Fig. 9. A major dust emission event occurred over the GD on 3 May (Fig.

9c). Shao2011 well simulated the dust emission event over the GD on 3 May (Fig. 9m), while GOCART obviously

underestimated dust emission over the GD (Fig. 9h). On 4 May, emitted dust from the GD was transported to the northeast

China, and the highest AOD values for this case study were observed in the northern China (Fig. 9d). As the GD is the main

dust source region of this dust storm, Shao2011 correctly captured the emission phase of this dust event. Simulated AOD

values from the S11Z01 configuration produced the closest match to the observed daily MODIS AOD with respect to the

magnitude and spatial pattern (Fig. 9n and Figure S3). For a more quantitative comparison, Table 6 shows the correlation

coefficient (R) and root mean square error (RMSE) between the model and observed AOD for six experiments during 1-7

May. Overall, S11Z01 experiment gives a larger correlation coefficient and the RMSE is almost the same among different

experiments, the correlation coefficient is still lower than 0.5. The low correlation may partly come from the spatial and

temporal limitation of satellites and the difficulties to retrieve aerosol in the vicinity of clouds for satellites.

To evaluate the model performance in simulating the vertical profile of dust aerosol, we compared the extinction coefficient

from model and from CALIPSO (Fig. 10). Figure 10 shows the simulated and observed aerosol extinction profiles at 532 nm

at 18:00 UTC 4 May. The trajectory of CALIPSO passes the East Asia (Fig. 10d). All the six experiments show the similar

dust location in the atmosphere, which is consistent with the CALIPSO observation. However, the magnitude of dust

concentration differs substantially. The simulated extinction coefficients using GOCART dust emission schemes are

significantly underestimated compared to the CALIPSO observation (Fig. 10a,10b and 10c), while the modeled extinction

coefficients using Shao2011 dust emission scheme agrees better with observation though they are still underestimated (Fig.

10e,10f and 10g). Among all the six experiments, results from S11Z01 agree the best with observation.

In summary, both ground and satellite observations indicate that the S11Z01 experiment yields the best performance in simulating this dust storm. As we discussed in Sect. 3.2, the Z01 dry deposition scheme indeed has a better physical treatment and performs better than some other dry deposition schemes.

## 4    Summary and discussion

In this study, we analyzed the dust simulation sensitivity to different dust emission schemes and dry deposition schemes. In order to compare different dust emission schemes, the Shao2011 dust emission scheme has been implemented into the MOSAIC aerosol scheme in WRF-Chem v3.9. Six model experiments were conducted to simulate the dust storm in May 2017 over East Asia, with two dust emission schemes (GOCART and Shao2011) and three dry deposition schemes (BS95, PE92 and Z01). The simulation results of different experiments were evaluated against surface and satellite observations.

Our results show that dust loading is very sensitive to different dry deposition schemes. The relative difference of dust loading in different experiments range from 20%-116% when using different dry deposition schemes. The difference of dry deposition velocity in different dry deposition schemes comes from the parameterization of surface resistance, and difference in surface resistance mainly comes from the parameterization of collection efficiency from impaction and rebound effect. In addition, different dust emission schemes result in different spatial distribution of dust loading, as dust emission fluxes in dust source

regions differ substantially among different dust emission schemes, which is mainly attributed to differences in the threshold conditions for dust emission and in formulas and parameters for calculating dust emission flux. We noted that, the Shao2011 dust emission scheme is different among different WRF-Chem versions, and significant difference exist in the simulated dust emission fluxes between WRF-Chem v3.9 and WRF-Chem v3.7.1, which is mainly caused by differences in soil particle size distributions used in two versions (see Appendix B).

Compared with both surface $PM_{10}$ station observations and MODIS AOD, the Shao2011 dust emission scheme coupled with the Z01 dry deposition scheme produces the best simulation for the dust storm in East Asia. For PM10, S11Z01 experiment gives a larger R of 0.83 and the smallest RMSE of 82.98 of all the stations (Table 5). The spatial distribution of AOD during the

simulation period obtained by S11Z01 agrees the best with MODIS AOD (Fig. 9), with the largest R and a relatively small RMSE (Table 6). Our analysis indicates Z01 accounts for the rebound effect and $E_{IM}$ changes with LUC and therefore has a better physical treatment of dry deposition velocity than the two other dry deposition schemes. Previous studies have also shown that the Z01 scheme agrees better with measured dry deposition fluxes and dry deposition velocity (e.g., Zhang et al., 2012; Connan et al, 2018). The Shao2011 dust emission scheme has larger dust emission fluxes than GOCART dust emission scheme over the Gobi Desert, and the transport of dust emitted from the Gobi Desert is the most important source of dust weather in northern China (Chen et al., 2017b). Compared with daily MODIS AOD (Fig. 9), our results indicate that dust emission from Shao2011 is better for this dust event, in terms of dust spatial and temporal distributions. We note that our results are obtained from simulations of a dust storm over a short period, and longer simulations are desirable in the future to test whether the optimal scheme here still produces best simulations.

This study highlights the importance of dry deposition process in dust simulation. Future studies on dust simulation should pay attention to improve dry deposition schemes as well as the dust emission schemes. Additional field measurements of dry deposition process and comparisons with model results are required to reduce the uncertainties on dust simulation.

## Appendix A: Description of the Shao2011 dust emission scheme

Here we describe the Shao2011 dust emission scheme in more detail as a supplement to the Sect. 2.2.2 of this article. The total saltation flux Q in Eq. (4) is calculated as:

$$Q = \int_{d_1}^{d_2} Q(d) p_m(d) \delta d \tag{A1}$$

where $d_1$ and $d_2$ define the upper and lower limits of saltation particle size. $p_m(d)$ is the the minimally disturbed particle-size distribution. The saltation flux Q for each particle size $d$ is calculated as:

$$Q(d) = (1 - c_f) c_0 \frac{\rho_a}{g} u_*^3 \left(1 - \frac{u_{*t}}{u_*}\right) \left(1 + \frac{u_{*t}}{u_*}\right)^2 \tag{A2}$$

where $c_f$ is the fraction of vegetation cover, $(1-c_f)$ means the fraction of erodible surface area, $c_0$ is a coefficient. $u_{*t}$ is the threshold friction velocity, $u_*$ is the friction velocity. When $u_*$ is larger than $u_{*t}$, it calculates the dust emission flux. Before WRF-Chem v4.0, there is a bug in calculating the saltation flux $Q(d_s)$ in Shao2011. They miscalculate the last term as

$(1+\left(\frac{u_{*t}}{u_*}\right)^2$ ) in WRF-Chem codes (LeGrand et al., 2019). In WRF-Chem v4.0 and later versions, they fixed this bug and we also fixed this bug in our simulations.

The threshold friction velocity $u_{*t}$ is calculated as:

$$u_{*t} = u_{*t0}f_\lambda f_\theta \tag{A3}$$

where $u_{*t0}$ is the ideal threshold friction velocity when soil is dry, bare and free of crust and salt, $f_\lambda$ is the correction functions for surface roughness, $f_\theta$ is the correction functions for soil moisture. The ideal threshold friction $u_{*t0}$ is calculated as:

$$u_{*t0} = \sqrt{a_1 \frac{\rho_p}{\rho_a} gd + \frac{a_2}{\rho_a d}} \tag{A4}$$

where $a_1$ and $a_2$ are constant. $\rho_p$ and $\rho_a$ are particle and air density. d is the particle diameter.

The correction functions for surface roughness $f_\lambda$ is calculated as:

$$f_\lambda = [(1 - m\sigma\lambda)(1 + m\beta_0\lambda)]^{\frac{1}{2}} \tag{A5}$$

where m is a constant, $\sigma$ is the ratio of roughness-element basal area to frontal area, $\lambda$ is the frontal area index, $\beta_0$ is the ratio of the drag coefficient of an isolated roughness element on the surface to the drag coefficient of the substrate surface itself.

The mass fraction of free dust $\eta_{mi}$ is calculated as:

$$\eta_{mi} = \int_{d-\frac{\Delta d_i}{2}}^{d+\frac{\Delta d_i}{2}} p_m(d)\delta d \tag{A6}$$

where $p_m(d)$ is the minimally disturbed particle-size distribution, which is regarded as a composite of several log-normal distribution, $p_m(d)$ is expressed as:

$$p_m(d) = \frac{1}{d} \sum_{i=1}^{J} \frac{w_j}{\sqrt{2\pi}\sigma_j} exp\left(-\frac{(lnd - lnD_j)^2}{2\sigma_j^2}\right) \tag{A7}$$

soil samples collected from experiment sites are used to determine the particle size distribution.

**Appendix B: The Shao2011 dust emission scheme in different versions of WRF-Chem**

As we noted in Sect. 3.2, the Shao2011 scheme in different versions of WRF-Chem can produce significantly different dust emission fluxes. Here we document differences in Shao2011 among different WRF-Chem versions:

1. The first difference is $c_0$. $c_0$ is a coefficient used to calculate the saltation flux as in Eq. (A2). In versions before WRF-Chem v3.8, $c_0$ is equal to 0.5; in WRF-Chem v3.8 and later versions, $c_0$ is equal to 2.3 (Table B1).

2. The second difference is $\beta_0$. $\beta_0$ is a coefficient used to calculate the correction function for surface roughness $f_\lambda$ in Eq. (A5). In versions before WRF-Chem v3.8, $\beta_0$ is 90; in WRF-Chem v3.8 and later versions, $\beta_0$ is 200 (Table B1).

3. The third difference is caused by the minimally disturbed particle-size distribution $p_m(d)$ (see Eq. (A6)). $p_m(d)$ is used to calculate the free dust fraction $\eta_{mi}$ (see Eq. (A6)). Free dust fraction is the fraction of dust that has lower enough binding energy so that it can be easily lifted from the surface by either aerodynamic forces or mechanical abrasion (Shao, 2001). The $\eta_{mi}$ is used to calculate the dust emission rate in Eq. (4). 12 soil types are included in all WRF-Chem versions. In WRF-Chem v3.8 and later versions, each soil type has a corresponding $p_m(d)$ as listed in Table B1 from Shao et al. (2010); in versions before WRF-Chem v3.8, there are only four $p_m(d)$ as listed in Table 1 from Shao (2004) for 12 soil types (Fig. B1). For example, (f) sand and (g) loamy sand soil types use the same free dust fraction distribution in versions before WRF-Chem v3.8. As shown in Table B2, the loam and clay loam are the two soil types with the largest percentage, while the other soil types account for a very small percentage. From Fig. B1c and Fig. B1e, for loam and clay loam soil types, the free dust fraction is so small in the particle size range 0-10 um in WRF-Chem v3.8 and later versions, almost all close to 0; while in the versions before WRF-Chem v3.8, the free dust fraction is relatively high. In different WRF-Chem versions, the total saltation flux Q is the same, but dust emission flux $F(d_i)$ is different due to different free dust fraction (see Eq. (4)). With smaller free dust fraction, the dust emission flux is smaller in WRF-Chem v3.8 and later versions.

To examine the importance of these changes, we run four experiments to quantify the contribution of each factor (Table B3). For control run, $c_0$ is 2.3, $\beta_0$ is 200 and $p_m(d)$ has 12 distributions based on WRF-Chem v3.8 or later versions. For case1

experiment, $\beta_0$ is changed to 90, the one used in WRF-Chem v3.7.1 and all other parameters are kept the same as in control run. The dust emission of case1 is 1.35 times higher than the control run. For case2 experiment, $c_0$ is changed to 0.5, the one used in WRF-Chem v3.7.1, and all other parameters remain the same as in control run. The dust emission of case2 is twenty-one percent of the dust emission of the control run. For case3 experiment, $p_m(d)$ is adopted from WRF-Chem v3.7.1 and has four distributions, and all other parameters remain the same as in control run. The dust emission of case3 is 13 times higher than the control run. This indicates that the difference of dust emission between different versions of Shao2011 scheme is mainly caused by the change of $p_m(d)$. As $p_m(d)$ is determined by soil particle size distribution, this also highlights the need to improve the accuracy of soil texture.

We should mention that the Shao2011 dust emission scheme we used in this study is based on WRF-Chem v3.9 with the soil particle size distribution from WRF-Chem v3.7.1, which simulates better dust loading compared with observations. Compared with the original Shao2011 scheme in WRF-Chem v3.9, the total dust emission simulated in our experiments during 1-7 May is 13 times higher.

**Code availability**

The source code of WRF-Chem is available at http://www2.mmm.ucar.edu/wrf/users/download/get_sources.html (last access: 31 October 2019). The modified WRF-Chem v3.9 with Shao2011 dust emission scheme implemented in MOSAIC aerosol scheme is available upon request to the corresponding author.

**Data availability**

The 6-hourly National Center for Environmental Prediction Final (NCEP/FNL) Operational Global Analysis data at a resolution of 1°×1° can be obtained from: https://rda.ucar.edu/ (last access: 31 October 2019). The observed PM10 data is collected from the National air quality real time release platform at: http://106.37.208.233:20035/ (last access: 31 October 2019). The historical data of air quality used in this study can be downloaded from: http://beijingair.sinaapp.com/ (last access: 31 October 2019). Daily MYD08_D3 files from the MODIS onboard Aqua satellite can be obtained from

https://ladsweb.nascom.nasa.gov (last access: 31 October 2019).

**Author contributions**

MW and YZ conceived the idea and designed the model experiments. YZ performed the simulations, conducted the analysis and wrote the manuscript. CZ and SC provided guideline for the dust simulations in WRF-Chem and helped in the interpretation of the results. XH and ZL provided guideline for the WRF-Chem simulations and contributed to the model experiments design. YG helped in the PM10 data processing and usage. Everyone edited the manuscript.

**Competing interests**

The authors declare that they have no conflict of interest.

**Acknowledgements**

This work is supported by the Minister of Science and Technology of China (2016YFC0200503 and 2017YFA0604002) and by the National Natural Science Foundation of China (41575073, 41621005 and 91744208). This research is also supported by the Collaborative Innovation Center of Climate Change, Jiangsu Province. Chun Zhao is supported by the Natural Science Foundation of China NSFC (41775146) and the Fundamental Research Funds for the Central Universities. The numerical calculations in this paper have been done on the computing facilities in the High Performance Computing Center (HPCC) of Nanjing University.

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

**Table 1.** WRF-Chem configuration

| Atmospheric Process | WRF-Chem option | Namelist Variable | Option |
|---|---|---|---|
| Surface Layer physics | Noah land-surface model | sf_surface_physics | 2 |
| Soil map | USGS | num_land_cat | 24 |
| Boundary Layer Physics | YSU scheme | bl_pbl_physics | 1 |
| Longwave/Shortwave Radiation | RRTMG | ra_lw(sw)_physics | 4 |
| Cumulus Clouds | Grell-Freitas | cu_physics | 3 |
| Cloud microphysics | Morrison double-moment | mp_physics | 10 |
| Gas-phase/aerosol chemistry | CBMZ/MOSAIC 4-bin | chem_opt | 9 |





**Table 2.** Three dry deposition schemes

| Scheme | BS95 | PE92 | Z01 |
|---|---|---|---|
| $V_d$ | $V_d = V_g + \dfrac{1}{R_a + R_s + R_a R_s V_g}$ | $V_d = V_g + \dfrac{1}{R_a + R_S}$ | $V_d = V_g + \dfrac{1}{R_a + R_s + R_a R_s V_g}$ |
| $R_S$ | $R_s = \dfrac{1}{u_*(E_B + E_{IM})}$ | $R_s = \dfrac{1}{u_*(E_B + E_{IM} + E_{IN})R}$ | $R_s = \dfrac{1}{\varepsilon_0 u_*(E_B + E_{IM} + E_{IN})R}$ |
| $E_{IM}$ | $E_{IM} = 10^{-St}$ | $E_{IM} = \left(\dfrac{St}{0.8 + St}\right)^2$ | $E_{IM} = \left(\dfrac{St}{\alpha + St}\right)^\beta$ |
| $E_{IN}$ | $0$ | $E_{IN} = \dfrac{(0.0016 + 0.0061 z_0)d_p}{1.414 \times 10^{-7}}$ | $E_{IN} = \dfrac{1}{2}\left(\dfrac{d_p}{A}\right)^2$ |
| $R$ | $1.0$ | $R_1 = e^{-2\sqrt{St}}$ | $e^{-1.0\sqrt{St}}$ |
| $St$ | $St = \dfrac{u_*^2 v_g}{gv}$ | $St = \dfrac{\rho_p d_p^2}{9\mu d_c} u$ | $St = \dfrac{v_g u_*}{gA}$ (vegetated surfaces) <br><br> $St = \dfrac{v_g u_*^2}{gv}$ (smooth surfaces) |



**Table 3.** Model experiments and the corresponding model configuration in WRF-Chem.

| Experiment name | Dust emission scheme | Dry deposition scheme | aer_drydep_opt |
|:---:|:---:|:---:|:---:|
| GOBS95 | GOCART | BS95 | 1 |
| GOPE92 | GOCART | PE92 | 101 |
| GOZ01 | GOCART | Z01 | 301 |
| S11BS95 | Shao2011 | BS95 | 1 |
| S11PE92 | Shao2011 | PE92 | 101 |
| S11Z01 | Shao2011 | Z01 | 301 |





**Table 4.** Correlation coefficient (R) and root mean square error (RMSE) between WRF-Chem simulation and FNL reanalysis data for daily mean temperature at 2 meters, U component of wind, V component of wind and wind speed at 10 meters during the dust event time period over the whole domain. The last two rows show the R and RMSE for the time-averaged

temperature at 2 meters, U component of wind, V component of wind and wind speed at 10 meters from 1 May to 7 May.

| Day | R/RMSE | Temperature | U | V | Wind speed |
|---|---|---|---|---|---|
| 1 May | R | 0.99 | 0.86 | 0.85 | 0.75 |
| 1 May | RMSE | 1.32 | 1.51 | 1.59 | 1.60 |
| 2 May | R | 0.99 | 0.90 | 0.88 | 0.82 |
| 2 May | RMSE | 1.28 | 1.61 | 1.60 | 1.70 |
| 3 May | R | 1.0 | 0.91 | 0.90 | 0.84 |
| 3 May | RMSE | 1.22 | 1.60 | 1.64 | 1.76 |
| 4 May | R | 0.99 | 0.87 | 0.87 | 0.78 |
| 4 May | RMSE | 1.35 | 1.57 | 1.49 | 1.63 |
| 5 May | R | 0.99 | 0.88 | 0.87 | 0.80 |
| 5 May | RMSE | 1.23 | 1.49 | 1.44 | 1.57 |
| 6 May | R | 0.99 | 0.88 | 0.87 | 0.80 |
| 6 May | RMSE | 1.32 | 1.56 | 1.52 | 1.63 |
| 7 May | R | 0.99 | 0.89 | 0.82 | 0.79 |
| 7 May | RMSE | 1.37 | 1.42 | 1.30 | 1.39 |
| 1 May to 7 May | R | 1.0 | 0.90 | 0.86 | 0.82 |
| 1 May to 7 May | RMSE | 1.03 | 1.08 | 0.98 | 1.11 |

**Table 5.** Correlation coefficient (R) and root mean square error (RMSE) between the model and observations for $PM_{10}$ over five sub regions and for all the stations over whole China in Fig. 7 for six experiments listed in Table 3.


| Region | R/RMSE | GOBS95 | GOPE92 | GOZ01 | S11BS95 | S11PE92 | S11Z01 |
|--------|--------|--------|--------|-------|---------|---------|--------|
| TD | R | 0.64 | 0.53 | 0.59 | 0.34 | 0.37 | 0.37 |
| TD | RMSE | 79.61 | 91.91 | 106.61 | 124.25 | 119.54 | 115.68 |
| GD | R | 0.75 | 0.78 | 0.82 | 0.76 | 0.75 | 0.74 |
| GD | RMSE | 174.81 | 137.14 | 77.81 | 193.23 | 128.21 | 82.58 |
| NCP | R | 0.75 | 0.73 | 0.73 | 0.78 | 0.76 | 0.77 |
| NCP | RMSE | 231.2 | 221.05 | 197.43 | 189.08 | 164.25 | 107.20 |
| NEP | R | 0.62 | 0.63 | 0.58 | 0.70 | 0.68 | 0.68 |
| NEP | RMSE | 177.17 | 174.52 | 171.96 | 159.47 | 144.77 | 126.91 |
| YR | R | 0.45 | 0.42 | 0.43 | 0.67 | 0.61 | 0.61 |
| YR | RMSE | 105.96 | 105.97 | 93.97 | 94.07 | 93.79 | 69.94 |
| Total | R | 0.50 | 0.60 | 0.63 | 0.85 | 0.83 | 0.83 |
| Total | RMSE | 146.58 | 137.96 | 120.57 | 133.71 | 113.88 | 82.98 |


**Table 6.** Correlation coefficient (R) and root mean square error (RMSE) between the model and MODIS observation for AOD for six experiments over the 7-day simulation period 1-7 May, 2017.


| | GOBS95 | GOPE92 | GOZ01 | S11BS95 | S11PE92 | S11Z01 |
|------|--------|--------|-------|---------|---------|--------|
| R | 0.32 | 0.37 | 0.35 | 0.40 | 0.39 | 0.42 |
| RMSE | 0.46 | 0.46 | 0.46 | 0.46 | 0.47 | 0.47 |




**Table B1.** Differences in Shao2011 dust emission scheme between different WRF-Chem versions

| | Before WRF-Chem v3.8 | WRF-Chem v3.8 and later |
|---|---|---|
| $c_0$ | 0.5 | 2.3 |
| $\beta_0$ | 90 | 200 |
| $\eta_{mi}$ | 4 types | 12 types |




**Table B2.** Percentage of each soil type in the whole East Asia domain

| Soil type | Sand | Loamy sand | Sandy loam | Silt loam | silt | loam |
|---|---|---|---|---|---|---|
| percentage | 2.6% | 0.2% | 4.0% | 9.3% | 0 | 47.6% |
| Soil type | Sandy clay loam | Silty clay loam | Clay loam | Sandy clay | Silty clay | clay |
| percentage | 8.6% | 0 | 21.7% | 0 | 0.05% | 6.0% |





**Table B3.** Sensitivity of the simulated total dust emission from the Shao2011 to model parameters over the 7-day simulation period 1-7 May, 2017. The multiple of the dust emission of different cases is calculated with respect to the control run.

| | $C_0$ | $\beta_0$ | $\eta_{mi}$ | Dust emission (Tg) | Multiple |
|---|---|---|---|---|---|
| control run | 2.3 | 200 | 12 types | 1.35 | 1.00 |
| case1 | 2.3 | 90 | 12 types | 1.83 | 1.35 |
| case2 | 0.5 | 200 | 12 types | 0.29 | 0.21 |
| case3 | 2.3 | 200 | 4 types | 17.5 | 13.00 |

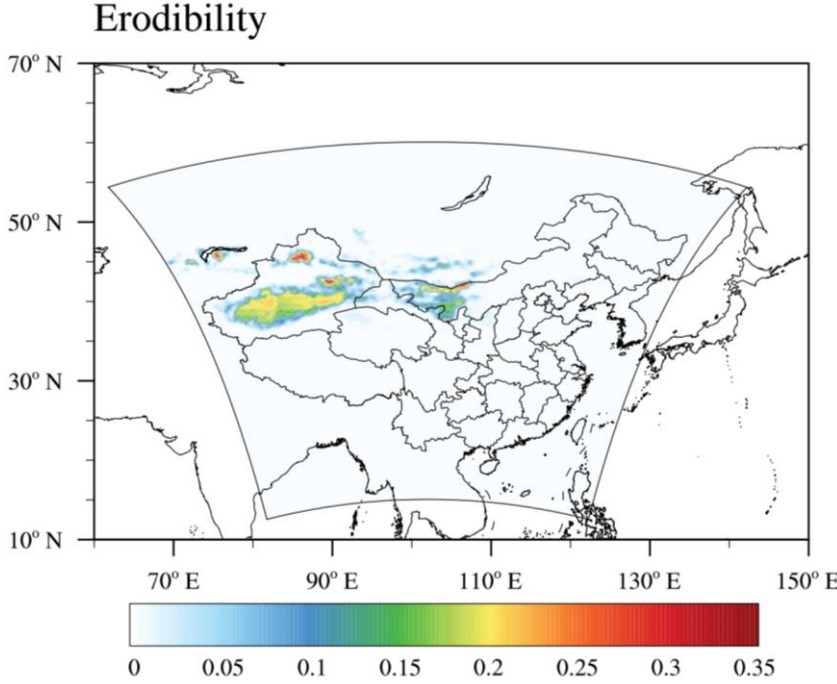

**Figure 1.** Domain map for the WRF-Chem simulations. The color shading shows the erodibility factor which is the fraction of erodible surface in each grid cell.

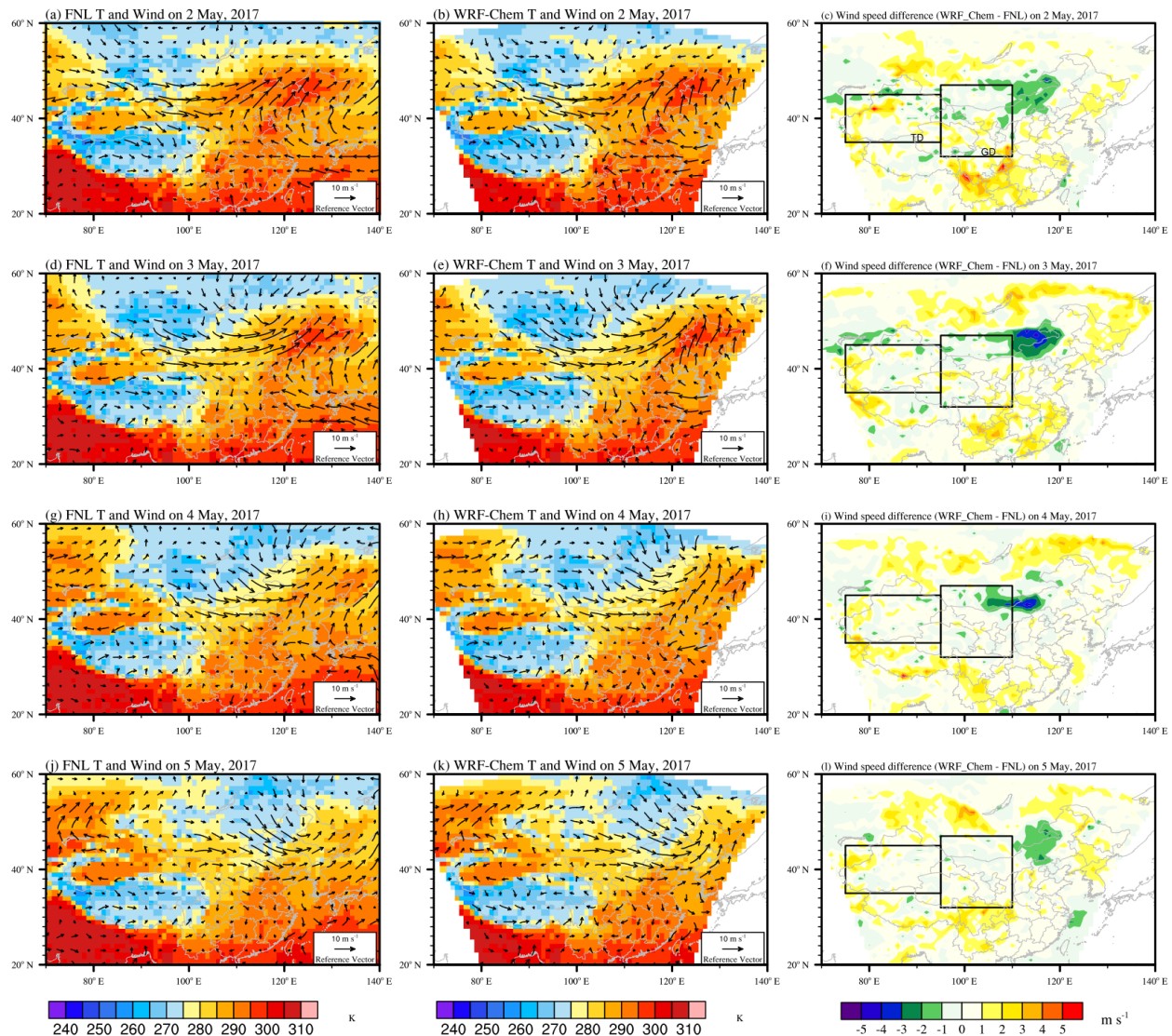

**Figure 2.** The left two columns show the surface meteorological conditions during the dust event. The color contours show the daily mean temperature field at 2 meters. Vectors represent the daily mean wind field at 10 meters. Panels (a)(d)(g)(j) show the NCEP/FNL reanalysis data. Panels (b)(e)(h)(k) show the WRF-Chem simulation. The rightmost column shows the difference of daily mean wind speed at 10 meters between WRF-Chem simulation and NCEP/FNL reanalysis data. The

rectangles show the dust source regions. "TD" is the Taklimakan Desert. "GD" is the Gobi Desert.

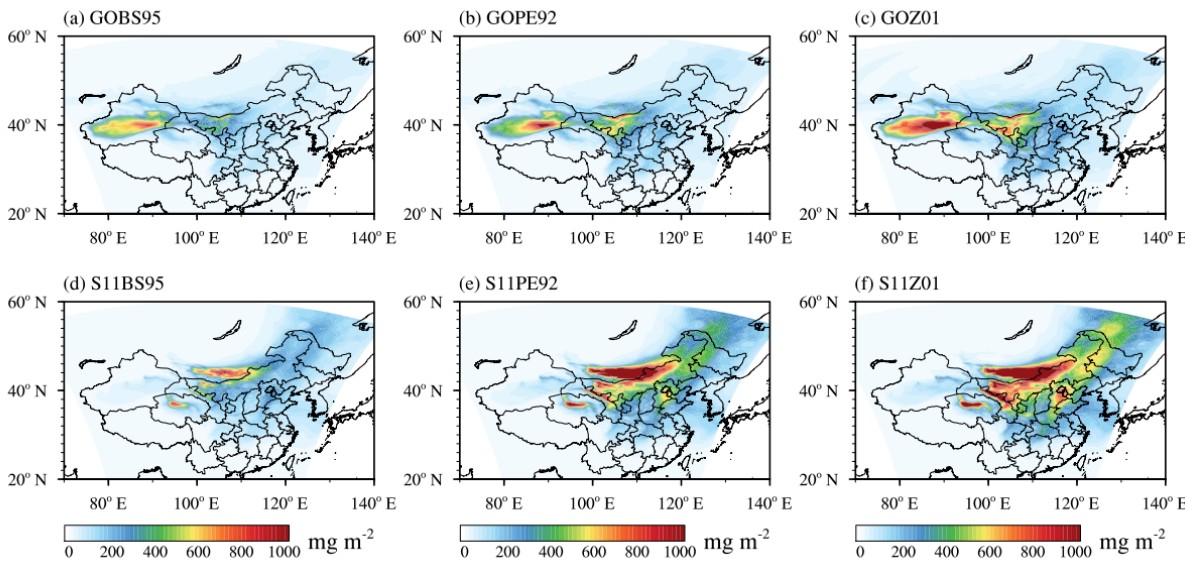

**Figure 3.** Spatial distribution of simulated mean dust loading for six experiments (a)GOBS95, (b)GOPE92, (c)GOZ01, (d)S11BS95, (e)S11PE92, (f)S11Z01 over the 7-day simulation period from 00:00 UTC on 1 May to 23:00 UTC on 7 May, 2017 (unit: mg m$^{-2}$).

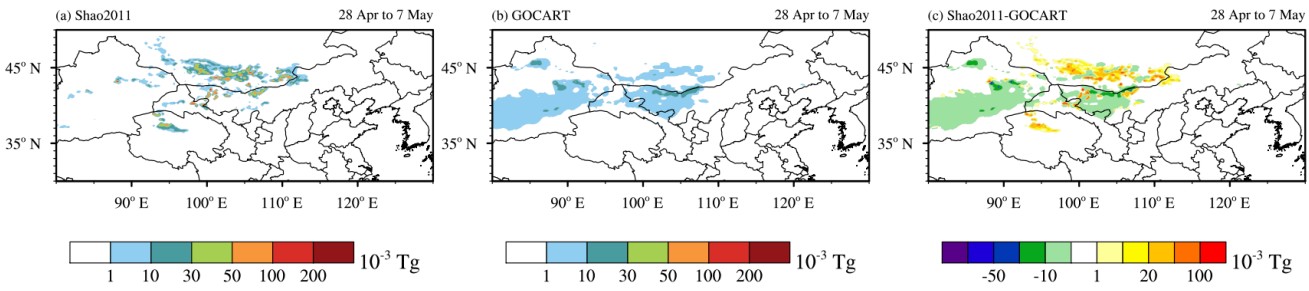

**Figure 4.** The simulated total dust emission ($10^{-3}$ Tg) from two dust emission schemes: (a) Shao2011 and (b) GOCART from 00:00 UTC on 28 April to 23:00 UTC on 7 May, 2017. (c) The total dust emission flux difference between Shao2011 and GOCART. The diameter of the emitted dust is less than 10 μm in both GOCART and Shao2011 dust emission schemes.

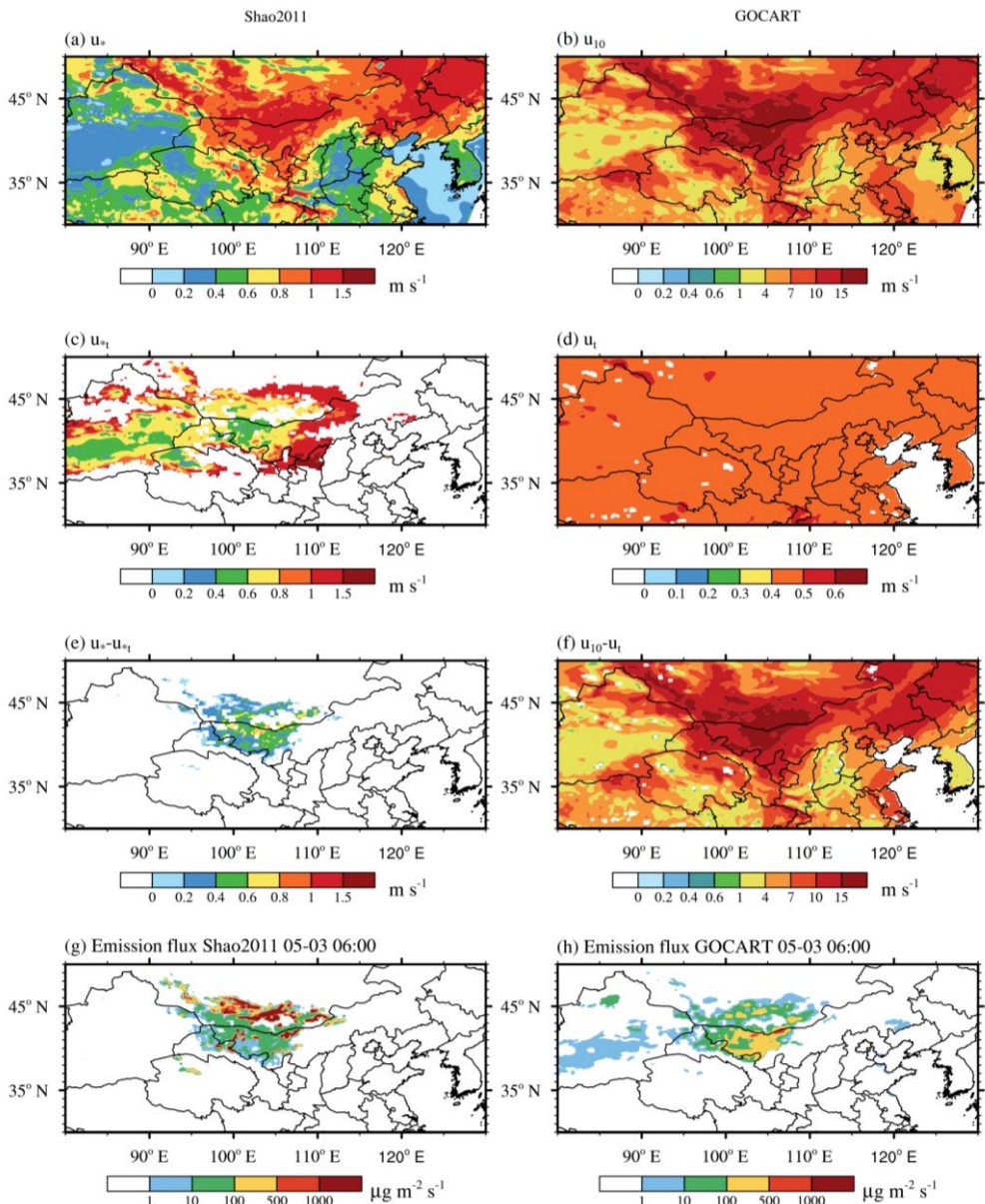

**Figure 5.** Spatial distribution of (a) friction velocity ($u_*$), (c) threshold friction velocity ($u_{*t}$) and (e) the difference between $u_*$ and $u_{*t}$ ($u_* - u_{*t}$) from Shao2011 dust emission scheme at 06:00 UTC on 3 May, 2017; (b) wind speed at 10 meters ($u_{10}$), (d) threshold velocity ($u_t$) and the difference between $u_{10}$ and $u_t$ ($u_{10} - u_t$) from GOCART dust emission scheme at 06:00 UTC on 3 May, 2017. Spatial distribution of (g) dust emission flux from Shao2011, (h) dust emission flux from GOCART at 06:00 UTC on 3 May, 2017.

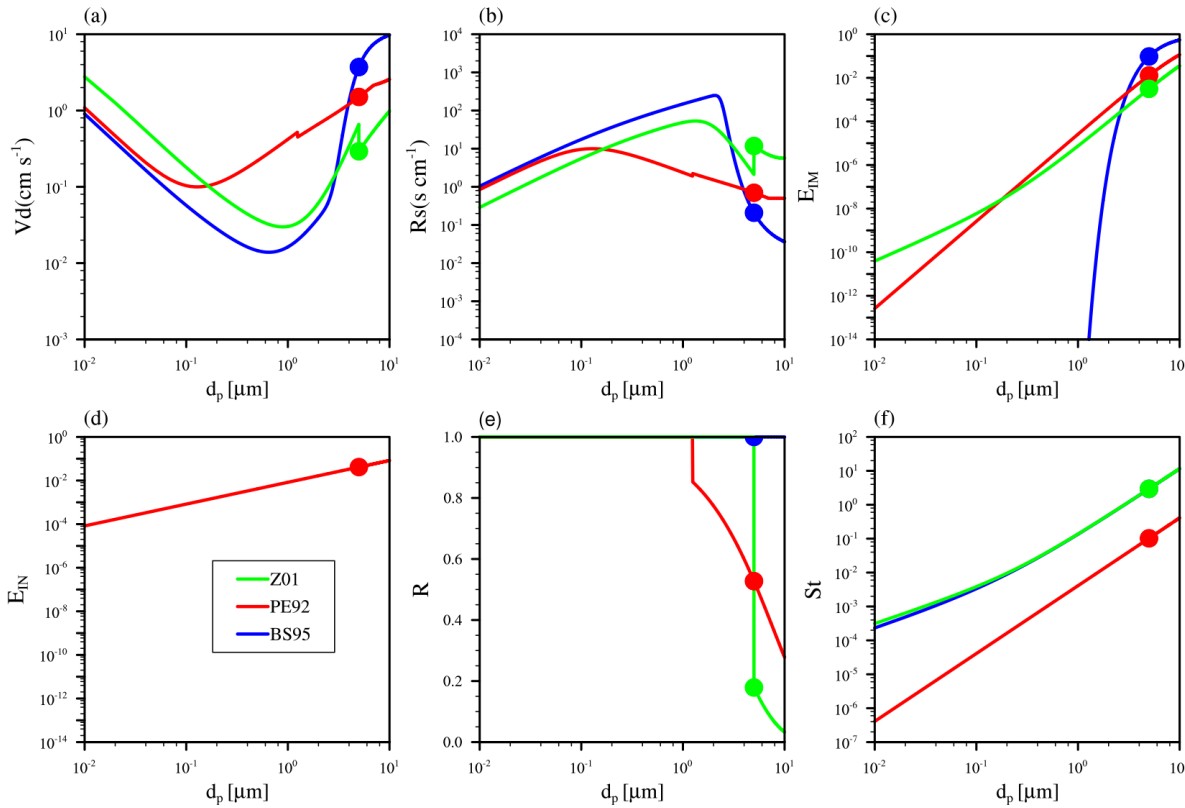

**Figure 6.** (a) Dry deposition velocity ($V_d$), (b) surface resistance ($R_S$), (c) surface collection efficiency from impaction ($E_{IM}$), (d) surface collection efficiency from interception ($E_{IN}$), (e) rebound (R) and (f) stokes number (St) as a function of particle diameter ($d_p$) over desert surface computed using different dry deposition schemes (BS95, PE92 and Z01). The colored dots indicate at the reference diameter of 5 μm.

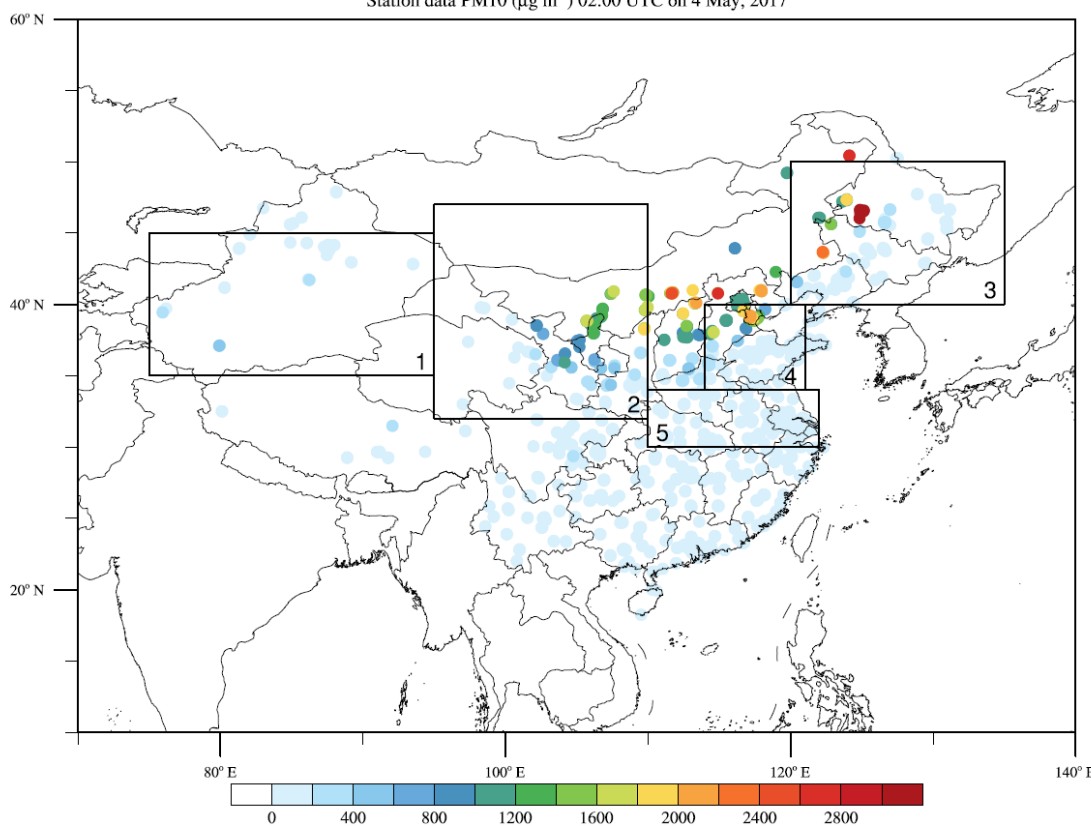

**Figure 7.** Five sub regions and observed $PM_{10}$ concentrations. "1" represents the Taklimakan Desert (TD), "2" represents the Gobi Desert (GD), "3" represents the Northeastern plain (NEP), "4" represents the North China Plain (NCP), "5" represents the Middle and lower reaches of Yangtze River Plain (YR). The colored dots represent observed $PM_{10}$ concentrations over observational sites at 02:00 UTC on 4 May, 2017 (10:00 Beijing Time (BJT) on 4 May, 2017).

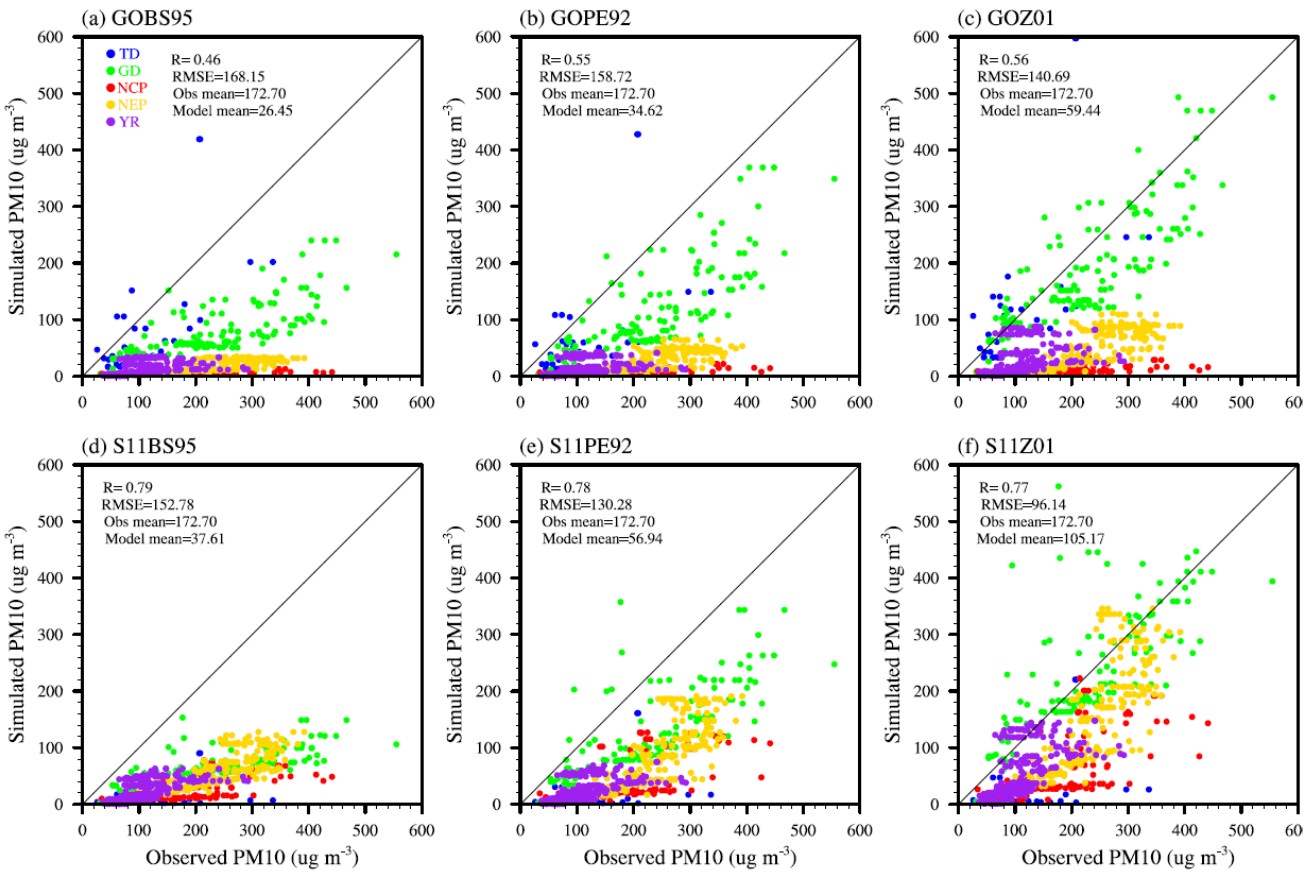

**Figure 8.** Simulated PM$_{10}$ versus observed PM$_{10}$ for six experiments (a)GOBS95, (b)GOPE92, (c)GOZ01, (d)S11BS95, (e)S11PE92, (f)S11Z01 over the 7-day simulation period 1-7 May, 2017. "Obs mean" is mean PM$_{10}$ over 1-7 May from observation, "Model mean" is mean PM$_{10}$ over 1-7 May from simulation, "R" is the correlation coefficient between model and observations, "RMSE" is the root mean square error. Different color dots represent different regions as shown in Fig. 6.

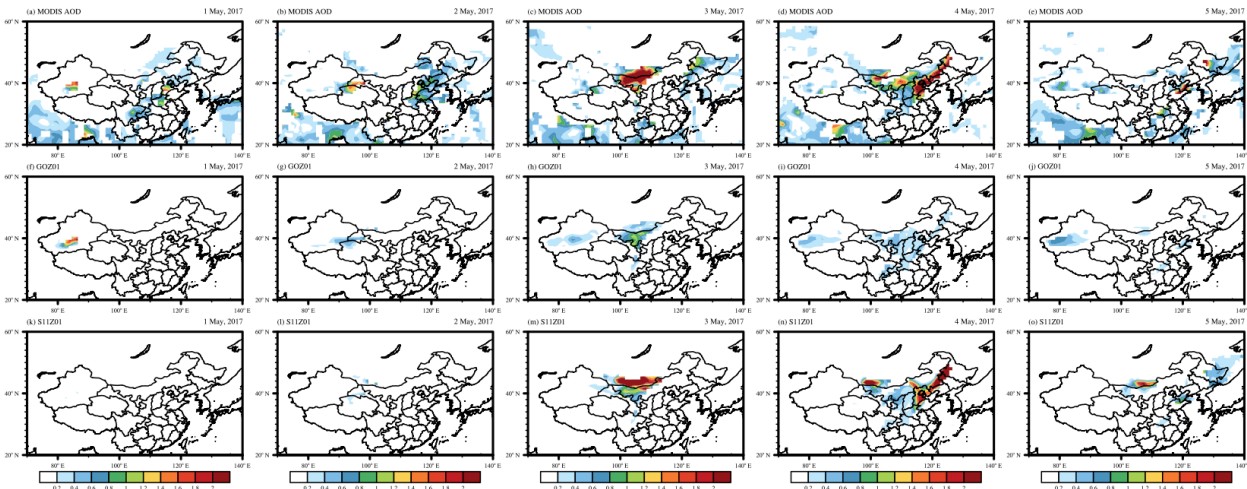

**Figure 9.** Simulated and observed mean AOD over the simulation period 1-5 May. Panels (a)(b)(c)(d)(e) show the distribution of daily mean aerosol optical depth (AOD) at 550 nm derived from MODIS-Aqua. Panels (f) (g) (h) (i) (j) show the WRF-Chem simulated AOD for GOBS95 experiment. Panels (k)(i)(l)(m)(n) show the WRF-Chem simulated AOD for

S11Z01 experiment. The model results are extracted from the simulation results at 13:00 local time for each region to match the MODSI observation time (details see Sect. 2.5). All the other experiments and for the period 1-6 May are shown in the supplementary (Fig. S1). Grid points without valid MODIS AOD retrieval are masked for both observational and model results.

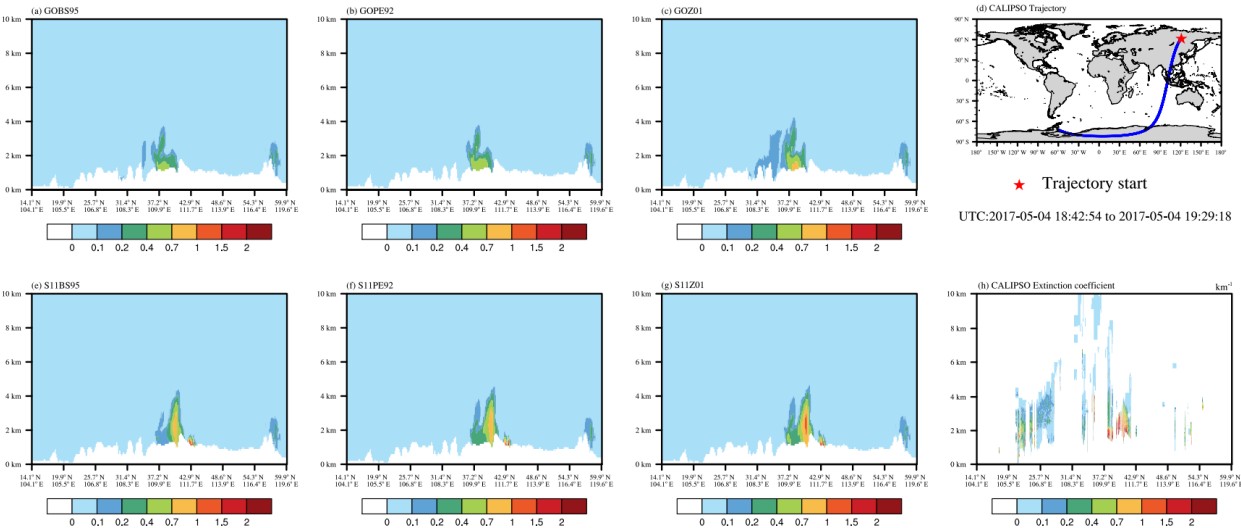

**Figure 10.** Simulated and observed aerosol extinction profiles at 532 nm at 18:00 UTC 4 May. Panel (d) show the CALIPSO

trajectory. Panel (h) show the CALIPSO observed extinction coefficient. Panels (a)(b)(c)(e)(f)(g) show the WRF-Chem

simulated extinction coefficient from six experiments.

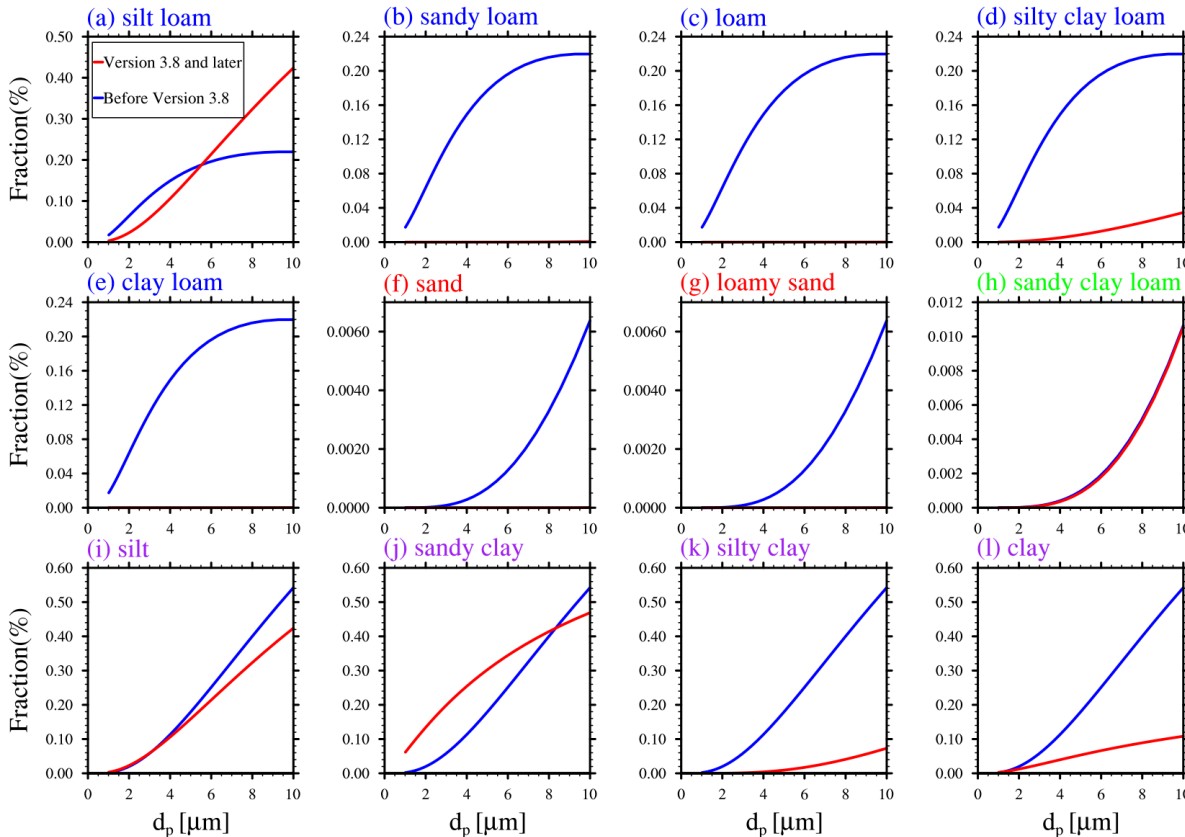

**Figure B1.** Free dust fraction for 12 soil types as a function of particle diameter ($d_p$). The red lines represent the free dust fraction in WRF-Chem v3.8 and later versions. The blue lines represent the free dust fraction before WRF-Chem v3.8. The colors of the soil type font in the upper left corner of the plot are different. In WRF-Chem v3.8 and later versions, each soil type has a corresponding free dust fraction distribution. In versions before WRF-Chem v3.8, several soil types share a free dust fraction distribution. The same soil type font color indicates that a free dust fraction is shared among these soil types in versions before WRF-Chem v3.8.