# Peer review of "WRF-Chem v3.9 simulations of the East Asian dust storm in May 2017"

_Geoscientific Model Development, 2019_

## Referee Comment (RC1) · Sagar Parajuli (Referee) · 15 Dec 2019

To the Editor: I find it very difficult to read the paper with the figures in the end of the paper. I do not understand why journals still like to see the figure in the end. These days, most people read the paper and do the review online. While reviewing online, it is inconvenient to go and see the corresponding figure by scrolling all the way to the end of the paper, and then come back again to continue reading the text. The figure should be placed inline with the text, near the text when the Figure is mentioned first. Anyway, this is an editorial issue and I am not sure how we can change this system.

Detailed Review: The paper is mostly well written and well structured. The main sci-

entific value of the paper is that it helps to identify the best depositional scheme for WRF Chem model. However, the model simulations were performed over a short period so the conclusions are questionable. Line 215: cite the figure that shows the study domain. 219: You say here: The meteorological conditions are reinitialized every 24 hours. But then you say only chemical conditions are reainitialized. Which one is true? Please specify the details explaining how exactly you implemented this in WRF Chem and which variables you updated. MOSAIC 8 bin scheme is computationally intensive and it is understandable why the authors chose 4 bin option. But I don't understand why you conducted simulations only for a dust episode (1week). Whenever one intends to evaluate a model, he/she must design the experiment more consciously. A year-long simulation is ideal because seasonal aspect has to be covered. This could be achieved by increasing the resolution or reducing the size of the domain. Of course, we must find a balanced model set up that serves to investigate our research goal. 245-285: These results are just some comparison and are not of much scientific value. There are several other previous studies which have compared different dust emission schemes. You have compared the simulated AOD with MODIS AOD which is good enough. Dust depositional aspect is not covered in the literature much so I suggest the authors to stick to the depositional aspect on this paper. I don't see why it was necessary to conduct simulations with different dust schemes because the main purpose of the study was to investigate the dry deposition processes. 200: It might be wise to define rebound effect, interception and collection efficiency in a simple, understandable language for the benefit of readers. Coming to the world of 'reality' from the world of 'equations' and giving some practical definition would be good. What happens when the dust falls on the leaves or a surface? What is the effect of type/condition of surface on dust deposition? Wet surface or dry surface, does it matter? And what happens after deposition? Does the deposited dust get blown away, or does it get washed away? Are these equations of depositions considering realistic processes? Or Are they just some 'theoretical' equations? 232: ….stations…please mention "locations shown in results section". 403-404: On what basis? Based on Table 4? Please refer to the

correlation/rmse values and discuss. 845: Figure 6, you mentioned that 1000 stations data were used but in the figure, it appears that most stations lie outside the six boxes chosen. Why not use all the stations? 850 Figure 7: Are the statistics calculated using daily-mean data? Please mention about this in the Figure caption. Line 368-373: The study domain is big and there exist time differences in different areas. How did you extract output at 13:00pm local time? If you are using level-3 MODIS data, using daily-average data would be fine because 1:30 pm local time is only at equator. Figure 8, the shaded color appears only in northern China region and data look empty in most region. Why so much data gaps in the model results? And why only use the data for May 7? My suggestion is to use time-averaged AOD during the dust episode and do the comparison. 380-384: Low correlation is understandable because WRF Chem can't reproduce dust events at hourly or daily time scale. It is extremely challenging to model short-lived dust events. So don't blame it to MODIS data. But, it would be better to time-average the data during the whole period (1-7may) and calculate spatial correlation coefficient and corresponding RMSE. Were R and RMSE calculate in this manner?

---

## Referee Comment (RC2) · Sandra LeGrand (Referee) · 19 Dec 2019

General Comments:

This study presents an investigation of dust simulation sensitivity to three dry dust deposition schemes and two adaptations of widely used dust emission schemes using the Weather Research and Forecasting model coupled with chemistry (WRF-Chem). The authors successfully demonstrate that airborne dust concentration and transport simulation can be sensitive to dry deposition process parameterization. Moreover, their findings make a compelling case that future efforts should focus on improving dry dust deposition schemes (in addition to dust emission schemes) and that more field measurements of dry deposition are needed to reduce uncertainties in dust simulation.

The authors did a good job introducing and comparing the deposition scheme physics in section 2.3 and should be commended for finding/reporting several undocumented discrepancies in the various WRF-Chem model versions. Overall, this paper brings attention to a process that is often overlooked in dust transport model assessment and is of value to the modeling community.

*However, critical gaps remain in the authors' methodology* that need to be addressed before publication moves forward.

The current approach supports the authors' assessment that GOCART and Shao2011 produce markedly different dust emission flux patterns, BS95 removed the most dust from the atmosphere, Z01 removed the least dust from the atmosphere, and the S11Z01 combination produced the best simulation of average $PM_{10}$ and AOD for this case study. These results, however, may only be applicable to this case study and particular WRF-Chem configuration.

For example, the authors did not include or allude to an analysis of how well the model simulated the general meteorological conditions driving the dust events. How well did the model winds (surface and aloft), synoptic conditions, etc. verify against observations? This is a necessary step to be able to discern whether simulation outcomes (good or bad) are actually due to dust scheme physics or an artifact of erroneous forcing conditions.

Also, it is unclear which dust emission scheme correctly captured the emission phase of this dust event (with respect to magnitude, spatial footprint, and temporal patterns). Assuming the Taklimakan and Gobi Deserts are the primary sources of dust, the scatter plots from Figure 7 seem to indicate GOCART did a better job with dust emission from the Taklimakan Desert and that there's little difference between the results of the two dust emission schemes for the Gobi Desert region. Figure 2 shows maps of the total simulated dust loading from the two dust emission schemes. GOCART clearly

produces widespread low-level dust emission, while Shao2011 emits stronger dust plumes from localized sources, mostly from the Gobi Desert region. The AOD comparison in Figure 8 is for midway through the dust event. It is unclear if dust originated in the Taklimakan Desert and was transported, if the Taklimakan Desert region in the AOD observation is cloud obscured, or if the dust in the Gobi Desert is entirely local.

Furthermore, because the authors have chosen to evaluate the dust emission and deposition schemes simultaneously, it's difficult to draw more generalizable conclusions about deposition scheme performance. Z01 may appear to be the best dry deposition scheme, but the slower deposition rate in Z01 may be compensating for the dust emission schemes not producing enough dust in the first place, issues with boundary layer mixing, or simulated winds that were too weak.

An overview of how well the model (or each simulation depending on whether or not aerosol feedbacks were affecting weather evolution) captured the general atmospheric conditions of the case study event is needed. This could be part of the main text or added as an appendix. This is particularly important given the strong influence of wind flow/turbulence and boundary layer mixing on the deposition process.

Dust emission observations are difficult to obtain (or in some cases non-existent). Understanding the evolution of the weather forcing conditions in combination with the dust emission simulation results, and possibly even qualitative assessment of true or false-color satellite imagery, would enable the authors to make inferences as to which dust emission treatment was more accurate for this particular case study. Timeseries plots comparing $PM_{10}$ observations to simulated $PM_{10}$ values from grid points in/near the source regions may also offer some insight.

Given the focus on deposition, this paper really should include an assessment of the simulated vertical dust distribution. For example, the authors could add a comparison of simulated vertical dust distribution to CALIOP LiDAR observations from the CALIPSO satellite (Winkler, 2009; available via the NASA Earth Data Portal at

https://search.earthdata.nasa.gov/) in order to demonstrate that simulated dust was in good agreement with observed plume heights before making assumptions about fall rate accuracy.

Hopefully, these issues can be easily addressed with additional plots and documentation. Papers by Ma et al. (2019), Letcher and LeGrand (2018), Rizza et al. (2017), and Nguyen et al. (2019) offer good examples of approaches for general dust case study descriptions, forcing weather evaluations, and/or vertical dust distribution assessments.

Specific Comments:

1. P3 L59: "Large-size" is a relative term. Please provide a value for a frame of reference. For example, "... large-size aerosol particles (e.g., diameters > X µm), such as dust."

2. P3 L69, P13 L282, P17 L398, P18 L401: "A lot" is somewhat colloquial for use in an academic paper.

3. P3 L71: Please adjust the text to make it clear that the papers by Yuan et al. (2019) and Chen et al. (2017) are also WRF-Chem studies.

4. P4 L75: Why is it important to evaluate the dust emission and deposition schemes simultaneously? Wouldn't it be better to select a case study with well-simulated dust emissions from a single dust emission scheme when assessing model sensitivity to deposition scheme configuration? Model performance assessment for different pairings of dust emission and deposition schemes over an extended period of time may be of value to some readers, but evaluating the two aspects of the dust transport process simultaneously for a single case study event introduces extra degrees of freedom that make it difficult to ascribe model performance to a particular root cause.

5. P4 L82-84: The authors reference a study by Zhang et al. (2019) that found that the WRF-Chem GOCART model underestimated dry deposition in northwest China by more than an order of magnitude compared to observations. Interestingly, the study by Zhang et al. (2019) was done using WRF-Chem v3.7.1. An error was recently discovered in how the GOCART gravitational settling code was implemented in WRF-Chem that also affects the calculation of the dry deposition rate (see code commit change comment in the WRF source code repository by Alexander Ukhov; https://github.com/openwfm/WRF-Fire-merge/commit/2ffdebf4ac311a5b1ef8cd0c639e0d857b550fdb). Given that this error wasn't corrected until the release of WRF-Chem v4.1, the findings from Zhang et al. (2019) may no longer be representative of GOCART in the current WRF-Chem release. It would be good if the authors note that here for reader awareness.

6. P5 L101: "The model setups are listed..." wording is odd. Suggest changing to "A summary of the settings used to configure the model are listed..."

7. P5 L103 and Table 1 – Please add the radiation time step to your model description. Simulated wind speeds and dust emission flux appear to be very sensitive to this parameter when using RRTMG (not well documented). Also, please include the land use dataset (*lu_index*) used for this study in the configuration description given that some of the deposition scheme parameters have dependencies on land use categories.

8. P5 L107-109: Suggest combining the following sentences to avoid redundancy: "The MOSAIC aerosol scheme uses sectional approach to represent aerosol size distribution. The MOSAIC 4-bin aerosol scheme divides aerosol particles into four size bins by aerosol diameter: 0.039-0.156, 0.156-0.625, 0.625-2.5, 2.5-10.0 µm." Suggest changing to: "The MOSAIC 4-bin aerosol scheme divides airborne particles into four size bins by their effective diameter (0.039-0.156, 0.156-0.625,

0.625-2.5, 2.5-10.0 µm) to represent aerosol size distribution."

9. P5 L112: Suggest deleting "from dust emission schemes" to avoid redundancy.

10. P6 L122-123: This statement as written (also stated in other peer-reviewed publications) misrepresents the findings from the Shao et al. (2011) paper. Shao et al. (2011) concluded that their simplified scheme produced similar results to the Shao (2004) scheme when compared to observations from the Japan-Australian Dust Experiment (JADE). The same Shao et al. (2011) paper also notes that these findings shouldn't be generalized due to the conditions of the JADE experiment. Recommend the authors simply note that they chose to use the most simplified version of the University of Cologne (UoC) dust emission schemes for their experiment or confirm Shao2004 and Shao2011 produce similar dust emission flux outcomes for their particular case study.

11. P6 Section 2.2.1: The GOCART emission scheme description is rather sparse compared to the Shao scheme descriptions. The authors reference the paper on MOSAIC by Zhao et al. (2010), which offers a similar brief overview of the GOCART dust emission scheme and references the original Ginoux et al. (2001) paper. However, closer examination of the code (subroutine *mosaic_source_du* in *module_mosaic_addemiss.F*) indicates *dust_opt=13* (at least in v3.9) also includes the modifications to the original GOCART dust emission scheme (*dust_opt=1*) documented by LeGrand et al. (2019; Section 3.2.1) with the exception of the $C$ parameter (default $C$ value is set to $1 \times 10^{-9}$ kg s$^2$ m$^{-5}$ consistent with the original Ginoux et al. (2001) paper). It would be good if the authors could expand this section given the general lack of documentation on the WRF-Chem *dust_opt=13* setting (currently not included in the WRF-Chem user's manual). Also, the paper by Zhao et al. (2010) explored more than one modal size distribution configuration. It would be beneficial to readers for the authors to describe how the emitted dust particle size distribution used in *dust_opt=13* is

prescribed.

12. P6 L142: ". . . we cut the size bins for MOSAIC aerosol scheme from Shao2011 directly." Please be clearer on how the emitted dust size bins were configured for both the GOCART and Shao2011 simulations. The UoC emitted dust size bins have diameter ranges of < 2.5, 2.5–5, 5–10, 10–20 μm in v3.7.1 and 0.2–2, 2–3.6, 3.6–6, 6–12, 12–20 μm in v3.9. UoC emitted dust size bins from v3.9 match the emitted dust size bins from GOCART *dust_opt=1*. Emitted dust size bins in GOCART *dust_opt=13* appear to be modified to ignore dust particles larger than 10 μm and match the 4-bin distribution used by MOSAIC (MOSAIC bins also noted by authors on P5 L109). Does this statement imply the authors modified the MOSAIC module aerosol size bins to incorporate 5 bins and larger particles (particles up to 20 μm) for the simulations configured with Shao2011? Also, the use of the word "cut" here is a little colloquial.

13. P7-10 Section 2.3: Please introduce what is meant by important terms like "rebound effect," "collection efficiency from interception," "Schmidt number," and "Stokes number" to help readers understand why differences in these parameters matter for the deposition process.

14. P9 L190 and L196: Please provide ranges for the $\alpha$ and $A$ parameters. Does use of Z01 have a dependency on WRF-Chem being configured with a particular land use dataset given the dependency pf $\gamma$, $\alpha$ and $A$ on the land use category (LUC)?

15. P10 Section 2.4: Please include the number of vertical levels used and the time step in the model description section.

16. P10 L214-215: This section needs a figure showing the model domain or a reference to one of the other figures showing the whole model domain. Suggest expanding Figure 4 (WRF-Chem EROD parameter) to include the whole model

domain, changing Figure 4 to Figure 1, and referencing the EROD parameter in
the dust emission scheme description section.

17. P10 L219-221: The authors note that meteorological conditions are reinitialized
every 24 hours, provide two examples of studies that also used this approach,
and comment that this approach has been verified to obtain better meteorological
fields. The references provided, however, don't really support this statement.
For example, Su and Fung (2015) reinitialize their meteorological fields ever 4
days, and neither study explored the use of different "spin-up" approaches on
their results. Reinitialization or "daisy-chain" spin-up is a common practice used
by numerical weather modelers. As long as the resultant weather fields used
in the experiment were representative, the justification statement (L220-221) is
unnecessary.

18. P11 Section 3 (Results): MOSAIC incorporates aerosol feedbacks. The six tests
most likely were subject to different weather forcing conditions as the simulations
evolved. How notable were those differences?

19. P11 L241-242: The authors utilize AOD simulation results at 1300 local time to
compare with the daily MODIS AOD product. The actual model domain encom-
passes multiple time zones though (e.g., Fig. 1). Are the model values used for
the analysis based on the central point of the model domain (UTC + 8 hours),
or was there some other approach used to create a composite simulation prod-
uct? A comparison of a single simulation time period to the daily product may
be fine if the model correctly captured the timing of the forcing conditions. Was
this the case? If not, it may be better to compare the daily MODIS product to the
simulation time period that best matches the state of the atmosphere when the
observations were collected or use simulated daily averaged-AOD values for the
comparison.

20. P11-12 L245-256: Which dry deposition scheme was used for the dust emission

analysis? MOSAIC includes aerosol feedbacks, which could affect the surface winds driving the dust emission simulation. The authors state that they reinitialized the meteorological conditions every 24 hours, but that could still allow enough time for the forcing conditions to be affected. Were the wind fields the same in both emission scheme tests?

21. P12 L258: Why was this time period chosen? Is it the highest magnitude of dust emission for the simulation event?

22. P12 L263-264 and L268-269: Unless the authors have altered the code, Shao2011 as implemented in WRF-Chem uses the EROD parameter from the original GOCART dust emission scheme as a mask. Dust emission is permitted where the erodibility factor is greater than zero via a binary (0 or 1) multiplier (e.g., LeGrand et al., 2019; section 3.3; implemented in *module_uoc_dust.F*). Note, areas classified as zero in the default pre-calculated erodibility factor dataset in WRF-Chem over land are either relatively high points in the terrain (maximum elevation in the surrounding $10°\times10°$ area) or determined to have vegetation coverage according to a static 1987 annual average land cover dataset derived from $1°\times1°$ resolution AVHRR data (see Kim et al., 2013). Was this erodibility factor masking treatment included in the code implemented by the authors into MOSAIC? If not, this is an important distinction to document.

23. P12 L257-258 and P36 Figure 3: What grain size(s) were used to diagnose $u_{*t}$ and $u_t$ (Fig. 3c through 3f)? Are the dust emission fluxes presented in Fig 3. (g and h) representative of the emission flux for that grain size or the total dust emission flux?

24. P12 L269-270: The authors' comment that differences in dust emission flux produced by the two dust emission schemes are due to differences in threshold conditions required for dust emission and differences in formulas and parameters used for calculating dust emission. In other words, the two dust emission

schemes are very different from each other and produce different results? This has been well documented in other publications and doesn't add to the discussion. Suggest removing this sentence. The narrative flows into the next paragraph without it.

25. P12 L272-275: GOCART is also dependent on mean wind shear. Intermittent turbulence is not considered in the GOCART dust emission process either. Dust emission under low wind speed in GOCART from the Taklimakan Desert is likely due to the threshold velocity error described in LeGrand et al. (2019). The erodibility factor values in the authors' model domain max out at 0.35. The application of the erodibility factor decreases the dust emission flux.

26. P13 L280-285: This is an important aspect of the experimental design and needs to be moved to the methodology section/incorporated into Section 2.2. Details about the Shao2011 configuration used in this study should be consolidated to Section 2.2; they should not split between the main text and the appendix.

27. P13 L294-295: Please provide a reference for the statement "As desert dust mass is mainly concentrated in the large particle size range..." and the upper bound of the range. Is this statement appropriate for all desert regions or just East Asia? Also please include the value of the reference diameter (5 µm) in the main text as well as the figure caption. The "coarse" and "accumulation" characterization of emitted dust from MOSAIC needs to be described prior to this discussion. These are somewhat ambiguous terms in the dust literature. Suggest defining these terms in section 2.2.

28. P15 L350: Suggest replacing the phrase "better than" with "more physically meaningful" here (also in the abstract). "Better physics" does not always translate to better numerical model simulations.

29. P16 L373: "Extremely high AOD" is a little too vague here. Are the observed AOD

values considered extremely high for this type of event in East Asia? Suggest replacing the sentence intro with "The highest AOD values for this case study were observed in..."

30. P17 L376-377: Wording here is a little odd. Suggest changing to "Simulated AOD values from the S11Z01 configuration produced the closest match to the observed daily MODIS AOD with respect to magnitude and spatial pattern (Fig. 8g).

31. P17 L382-383: Was the cloud cover an issue? Were there any areas masked out for clouds in the MODIS AOD observations that may have actually been high in dust concentration?

32. P18 L408-409: Reference needed for the statement "... dust emitted from [the] Gobi Desert is the most important source of dust weather in northern China."

33. P18 L409-410: The paper by Su and Fung (2015) provides an analysis of a single case study event in East Asia. This study offers valuable information, but a single case study is not sufficient evidence to make general claims about model performance over a region. Recommend removing the statement about the Shao2011 scheme being documented to give better performance than GOCART over East Asia from the text. It's unnecessary.

34. P19 L424: Why is there an ellipsis (...) in Eq. A2?

35. P19 L431-432, P30 L446-447, L463-464, P33 Table B1, P35 Table B3: Suggest adding a subscript to $\beta$ from Eq. A4 since the $\beta$ symbol is also used to represent a different parameter in one of the deposition schemes.

36. P30 Table 3: Columns for *dust_opt* and *dust_schme* are unnecessary. The value used to activate the Shao2011 dust emission module by the authors may not be the one used by the WRF-Chem source code managers. Listing an arbitrary

setting in Table 3 could cause confusion to readers if this new approach is eventually implemented into the baseline code with different activation options later. Suggest noting that GOCART is *dust_opt=13* and that Shao2011 is *dust_opt=4* with *dust_schme=3* in the text in section 2.2 and removing these columns from the table.

37. P33 Table B1: The values for the third row seem to be missing.

38. P37 Figure 2: The diameter of the emitted dust is less than 10 μm in the *dust_opt=13* version of GOCART. Unless the authors have modified the code, the upper range of the emitted dust size bins from *dust_opt=3* is 20 μm (in both v3.7.1 and v3.9).

Noted Typos:

1. P14 L302 and P40 Fig 5: Be consistent with symbol case. The particle diameter is represented by a lower case $d$ in all previous equations.

2. P11 L241-242: Use of "p.m." is unnecessary with 24-hour clock time.

3. P12 L253: Use of acronym GD for Gobi Desert before it's been defined (on P13 L277).

4. P12 L254: Use of acronym TD for Taklimakan Desert before it's been defined (on P12 L271).

5. P16 L372: "...with MODIS [is provided] in Fig. 8."

6. Punctuation is an issue. Several commas missing from compound sentences throughout the text.

7. Missing the word "the" before desert names throughout the text.

**References**

Chen, S., Huang, J., Qian, Y., Zhao, C., Kang, L., Yang, B., Wang, Y., Liu, Y., Yuan, T., Wang, T., Ma, X. and Zhang, G.: An overview of mineral dust modeling over East Asia, J. Meteorol. Res., 31(4), 633-653, doi:10.1007/s13351-017-6142-2, 2017.

Ginoux, P., Chin, M., Tegen, I., Prospero, J. M., Holben, B., Dubovik, O. and Lin, S.-J.: Sources and distributions of dust aerosols simulated with the GOCART model, J. Geophys. Res., 106(D17), 20,255-20,273, doi:10.1029/2000JD000053, 2001.

Kim, D., Chin, M., Bian, H., Tan, Q., Brown, M. E., Zheng, T., You, R., Diehl, T., Ginoux, P. and Kucsera, T.: The effect of the dynamic surface bareness on dust source function, emission, and distribution, J. Geophys. Res.-Atmos., 118(2), 871-886, doi:10.1029/2012JD017907, 2013.

LeGrand, S. L., Polashenski, C., Letcher, T. W., Creighton, G. A., Peckham, S. E. and Cetola, J. D.: The AFWA dust emission scheme for the GOCART aerosol model in WRF-Chem v3.8.1, Geosci. Model Dev., 12(1), 131-166, doi:10.5194/gmd-12-131-2019, 2019.

Letcher, T. W. and LeGrand, S. L.: A Comparison of Simulated Dust Produced by Three Dust-Emission Schemes in WRF-Chem: Case Study Assessment, ERDC/CRREL TR-18-13, U.S. Army Engineer Research and Development Center, Hanover, New Hampshire, USA, doi: 10.21079/11681/28868, 2018.

Ma, S., Zhang, X., Gao, C., Tong, Q., Xiu, A., Zhao, H. and Zhang, S.: Simulating performance of CHIMERE on a late autumnal dust storm over Northern China, Sustainability, 11(4), 1074, doi:10.3390/su11041074, 2019.

Nguyen, H. D., Riley, M., Leys, J. and Salter, D.: Dust storm event of February 2019 in Central and East Coast of Australia and evidence of long-range transport to New Zealand and Antarctica, Atmosphere, 10(11), 653, doi:10.3390/atmos10110653, 2019.

Rizza, U., Miglietta, M. M., Mangia, C., Ielpo, P., Morichetti, M., Iachini, C., Virgili, S. and Passerini, G.: Sensitivity of WRF-Chem model to land surface schemes: Assessment in a severe dust outbreak episode in the Central Mediterranean (Apulia Region), Atmos. Res., 201, 168-180, doi:10.1016/j.atmosres.2017.10.022, 2018.

Shao, Y.: Simplification of a dust emission scheme and comparison with data, J. Geophys. Res., 109(D10), doi:10.1029/2003JD004372, 2004.

Shao, Y., Ishizuka, M., Mikami, M. and Leys, J. F.: Parameterization of size-resolved dust emission and validation with measurements, J. Geophys. Res., 116(D8),

doi:10.1029/2010JD014527, 2011.

Su, L. and Fung, J. C. H.: Sensitivities of WRF-Chem to dust emission schemes and land surface properties in simulating dust cycles during springtime over East Asia: Simulated Dust Cycles Over East Asia, J. Geophys. Res.-Atmos., 120(21), 11,215-11,230, doi:10.1002/2015JD023446, 2015.

Winker, D. M., Vaughan, M. A., Omar, A., Hu, Y., Powell, K. A., Liu, Z., Hunt, W. H. and Young, S. A.: Overview of the CALIPSO mission and CALIOP data processing algorithms, J. Atmos. Oceanic Tech., 26(11), 2310-2323, doi:10.1175/2009JTECHA1281.1, 2009.

Yuan, T., Chen, S., Huang, J., Zhang, X., Luo, Y., Ma, X. and Zhang, G.: Sensitivity of simulating a dust storm over Central Asia to different dust schemes using the WRF-Chem model, Atmos. Environ., 207, 16–29, doi:10.1016/j.atmosenv.2019.03.014, 2019.

Zhang, X.-X., Sharratt, B., Lei, J.-Q., Wu, C.-L., Zhang, J., Zhao, C., Wang, Z.-F., Wu, S.-X., Li, S.-Y., Liu, L.-Y., Huang, S.-Y., Guo, Y.-H., Mao, R., Li, J., Tang, X. and Hao, J.-Q.: Parameterization schemes on dust deposition in northwest China: Model validation and implications for the global dust cycle, Atmos. Environ., 209, 1-13, doi:10.1016/j.atmosenv.2019.04.017, 2019.

Zhao, C., Liu, X., Leung, L. R., Johnson, B., McFarlane, S. A., Gustafson, W. I., Fast, J. D. and Easter, R.: The spatial distribution of mineral dust and its shortwave radiative forcing over North Africa: modeling sensitivities to dust emissions and aerosol size treatments, Atmos. Chem. Phys., 10(18), 8821–8838, doi:10.5194/acp-10-8821-2010, 2010.

---

## Author Comment (AC1) · 18 Feb 2020

We are grateful to the referee #1 and referee #2 for the valuable comments, which helped us to further improve our manuscript. Below we address the reviewer's comments, with reviewer comments are in black and our answers are in purple.

Before we provide the detailed point to point reply, we provide an overview of main changes and improvements:

1. A new figure and a new Table are included (see Fig. 2 and Table 4 in the revised manuscript) to evaluate the meteorological conditions, and these are discussed in in the beginning of the results part. It shows that the WRF-Chem performed well in simulating the meteorological conditions during our simulation period. So, the differences of simulated dust emission and deposition are attributed to the different emission and deposition schemes.

2. We examined how the daily AOD evolves during the whole dust event period in Fig. 9. It is helpful to better quantify which dust emission scheme work better for this case, as this will further help to quantify which dry deposition schemes works better. Compared with daily MODIS AOD (Fig. 9), our results indicate that dust emission from Shao2011 is better for this dust event, in terms of dust spatial and temporal distributions. We can then comfortably conclude that Z01 dry deposition scheme performs the best among three dry deposition schemes we evaluated (Fig. 8).

3. The vertical extinction coefficient profile from CALIPSO is used to evaluate the vertical dust distribution (Fig.10).

4. We added more description for the important terms like "rebound effect," "collection efficiency from interception, "collection efficiency from impaction " to help readers understand why differences in these parameters matter for the deposition process (Sect. 2.3).

**Referee #1**
Detailed Review: The paper is mostly well written and well structured. The main scientific value of the paper is that it helps to identify the best depositional scheme for WRF Chem model. However, the model simulations were performed over a short period so the conclusions are questionable.

Thank you for the comments. As the main purpose of this paper is to highlight the importance of the dry deposition process that is often overlooked in dust transport models, even with a short time period dust storm simulation, we still can see the large dust loading difference which is attributed to the dry deposition scheme. What is more, we further evaluated simulated meteorological fields and found that simulated fields are close to those from the reanalysis fields and that different experiments produce very similar meteorological fields in the revised manuscript. This partly alleviates the limitation of the short-term simulation. We do acknowledge the potential limitation of the simulation over a short period, and longer simulations are desirable in the future to test whether the optimal scheme here still produces best simulations. This is now also added in Sect. 4.

1. Line 215: cite the figure that shows the study domain.

Thank you for the suggestion. Now we expanded Fig.4 to show the study domain and changed it to Fig.1. And we also cited the Fig.1 in Sect. 2.4 (P11 L253).

2. 219: You say here: The meteorological conditions are reinitialized every 24 hours. But then you say only chemical conditions are reinitialized. Which one is true? Please specify the details explaining how exactly you implemented this in WRF Chem and which variables you updated.

Thanks. The meteorological conditions are reinitialized but the chemistry not reinitialized. For meteorological conditions (such as wind speed and temperature), we reinitialized every 24 hours using NCEP/FNL reanalysis data. For chemistry, the output of the aerosol field (such as the concentration of different aerosol species) from the previous 1-day run was used as the initial chemical conditions for the next 1-day run. Our simulation period is from 26 April to 7 May 2017. So one experiment consists of 12 one-day run. For each one-day run, the FNL/NCEP data is used as initial meteorological condition, and the chemistry is from the last time step of the last one-day run. This can be achieved by setting chem_in_opt=1 in namelist.input. In this way, the chemical field are continuous and we can also get more reliable

meteorological conditions. We also specify the details in the main text ( P12 L259-263).

3. MOSAIC 8 bin scheme is computationally intensive and it is understandable why the authors chose 4 bin option. But I don't understand why you conducted simulations only for a dust episode (1week). Whenever one intends to evaluate a model, he/she must design the experiment more consciously. A year-long simulation is ideal because seasonal aspect has to be covered. This could be achieved by increasing the resolution or reducing the size of the domain. Of course, we must find a balanced model set up that serves to investigate our research goal.

Thank you for the comments. Please also see our reply to general comment from reviewer #1. As the main purpose of this paper is to highlight the importance of the dry deposition process that is often overlooked in dust transport models, even with a short time period dust storm simulation, we still can see the large dust loading difference which is attributed to the dry deposition scheme. What is more, we further evaluated simulated meteorological fields and found that simulated fields are close to those from the reanalysis fields and that different experiments produce very similar meteorological fields in the revised manuscript. This partly alleviates the limitation of the short-term simulation. Because we simulated the complete processes of a typical spring dust event in East Asia, it also has certain value for the prediction and simulation of dust storm cases in East Asia. We do acknowledge the potential limitation of the simulation over a short period, and longer simulations are desirable in the future to test whether the optimal scheme here still produces best simulations. This is now also added in Sect. 4.

4. 245-285: These results are just some comparison and are not of much scientific value. There are several other previous studies which have compared different dust emission schemes. You have compared the simulated AOD with MODIS AOD which is good enough. Dust depositional aspect is not covered in the literature much so I suggest the authors to stick to the depositional aspect on this paper. I don't see why it was necessary to conduct simulations with different dust schemes because the main purpose of the study was to investigate the dry deposition processes.

Thanks for your comments! The main purpose of this study is to study the sensitivity of dust simulation to dry deposition schemes, and the performance of different dry deposition schemes. Here are two reasons why we also want to evaluate dust

emission scheme. One is that although there have been several previous studies comparing different dust emission schemes (Kang et al., 2011; Su and Fung, 2015; Wu and Lin, 2014; Yuan et al., 2019), their studies are all under the framework of the GOCART aerosol scheme. The dry deposition treatment of GOCART aerosol scheme seems to have big problems for dust simulation (Zhang et al., 2019). And the size distribution of the emitted dust is also different between the GOCART and MOSAIC aerosol scheme. So we do not know yet which dust emission scheme is best for our dust storm simulation based on the previous studies. Another reason is that in the currently released version of WRF Chem, only GOCART dust emission scheme is coupled in MOSAIC aerosol scheme. As shown in our manuscript, the GOCART dust scheme strongly underestimated dust concentrations in comparison with observations, no matter which dry deposition scheme we used. Therefore, we have newly implemented the Shao2011 scheme into MOSAIC aerosol scheme. If we put the results of Shao2011 directly, the readers may wonder why we don't use the default scheme and make large efforts to implement a new one. If no comparison of these two dust emission schemes is included in our manuscript, readers may feel confused. Therefore, we also briefly compare the differences between the two widely used dust emission schemes, which can provide a reference for readers who want to use the MOSAIC aerosol scheme to simulate dust storm in the future.

5. 200: It might be wise to define rebound effect, interception and collection efficiency in a simple, understandable language for the benefit of readers. Coming to the world of 'reality' from the world of 'equations' and giving some practical definition would be good. What happens when the dust falls on the leaves or a surface? What is the effect of type/condition of surface on dust deposition? Wet surface or dry surface, does it matter? And what happens after deposition? Does the deposited dust get blown away, or does it get washed away? Are these equations of depositions considering realistic processes? Or Are they just some 'theoretical' equations?

We added the description for rebound effect in the main text and it reads "When large particles (usually >5 μm) hit the non-sticky surface, they are liable to rebound from the surface if they have sufficient kinetic energy. The rebound factor R represents the fraction of particles that stick to the surface." (P11 L242-244)
We added the description for collection efficiency from interception in the main text and it reads "$E_{IN}$ is the collection efficiency based on the relative dimensions of the particle to the collector diameter (Gallagher, 2002). Interception occurs when

particles moving with the mean flow and the distance between an obstacle and particle center is less than half of the diameter. Then the particles will collide with and be collected by the obstacle" (P10 L228-231)

We added the description for collection efficiency from impaction in the main text and it reads "$E_{IM}$ is the collection efficiency due to impaction of the particle with the collecting surface (Gallagher, 2002). Impaction occurs when there are changes in the direction of airflow, and particles that cannot follow the flow will collide with the obstacle and stay on the surface due to the inertia (Giardina and Buffa, 2018)." (P9 L194-196)

6. 232: . . ..stations. . .please mention "locations shown in results section".

Improved. (P13 L277)

7. 403-404: On what basis? Based on Table 4? Please refer to the correlation/rmse values and discuss.

We added the correlation/rmse values and discussed (P22 L506-509). It reads "For PM10, S11Z01 experiment gives the largest R of 0.83 and the smallest RMSE of 82.98 of all the stations (Table 5). The spatial distribution of AOD during the simulation period obtained by S11Z01 is closet to MODIS AOD (Fig. 9), with a largest R and a relatively small RMSE (Table 6)."

8. 845: Figure 6, you mentioned that 1000 stations data were used but in the figure, it appears that most stations lie outside the six boxes chosen. Why not use all the stations?

In Fig8 (Fig. 7 in the original manuscript), we want to analyze the performance of different experiments in different regions. So we divided the domain into five subregions, with two dust source regions and three remote regions that is largely affected by this dust storm. We did calculate the R and RMSE for all the stations in Table 5 (see the last row of Table 5). We note that the total is not for all the stations of the subregions but for all the stations over the whole China that we showed in Fig. 7 (more than 1000 stations), and this is now clarified in the header of Table 5 (P37 L842).

9. 850 Figure 7: Are the statistics calculated using daily-mean data? Please mention about this in the Figure caption.

Thanks. We used time-averaged PM10 data over 1-7 May. We also added this in the Figure 8 Caption (P49 986-987).

10. Line 368-373: The study domain is big and there exist time differences in different areas. How did you extract output at 13:00pm local time? If you are using level-3 MODIS data, using daily-average data would be fine because 1:30 pm local time is only at equator.

First, we divided the domain into different time zones according to the longitude. Then we can get the UTC when the local time is 13:00 in different time zones. Next, the simulated AOD results are extracted at the corresponding UTC to build an AOD map for the entire domain for each day. Finally, these processed daily model results are compared with the MODIS daily AOD data. For example, for the UTC+8 Time Zone with longitude near 120E, we use the WRF-Chem simulated AOD at 05:00 UTC to compare with the MODIS daily AOD data.
Yes, we are using the level-3 MODIS data. As the A-Train satellites pass most region of Earth at around 13:30 local time, we think it would be more reasonable to use the model results at 13:00 local time in each region to compare with MODIS daily AOD data.
We also revised our description of the comparison method to make it clear (P13 L289-292).

11. Figure8, the shaded color appears only in northern China region and data look empty in most region. Why so much data gaps in the model results? And why only use the data for May 7? My suggestion is to use time-averaged AOD during the dust episode and do the comparison.

As for data gaps in the model results, this is because the model results are already collocated with MODIS observation. Grid points without valid MODIS AOD retrieval are masked for both observational and model results in Fig. 9. In this way, it is more intuitive to see which experiment is closer to MODIS AOD. This question is mentioned in the caption of Fig. 9 (P50 L1003-1004). We also describe this in the main text. (P20 L458-459)

For the second question, as we also want to show the different stages of this dust storm (e.g. dust emission and dust transport), we now plot the AOD at 13:00 local time during this episode from 1-5 May (Fig.9). And the description has been updated accordingly (P20 L460-470).

12. 380-384: Low correlation is understandable because WRF Chem can't reproduce dust events at hourly or daily time scale. It is extremely challenging to model short-lived dust events. So don't blame it to MODIS data. But, it would be better to time-average the data during the whole period (1-7may) and calculate spatial correlation coefficient and corresponding RMSE. Were R and RMSE calculate in this manner?

Thanks. The R and RMSE are indeed calculated during the whole period (1-7 May) in Table 6 (Table 5 in the original manuscript). For each day, we just use the results at 13:00 local time for each region. Then we averaged the model results at 13:00 of the 7 days and compared them with MODIS 7-day average. Please see our reply to #9 for the detail calculation.

**Referee #2**
This study presents an investigation of dust simulation sensitivity to three dry dust deposition schemes and two adaptations of widely used dust emission schemes using the Weather Research and Forecasting model coupled with chemistry (WRF-Chem). The authors successfully demonstrate that airborne dust concentration and transport simulation can be sensitive to dry deposition process parameterization. Moreover, their findings make a compelling case that future efforts should focus on improving dry dust deposition schemes (in addition to dust emission schemes) and that more field measurements of dry deposition are needed to reduce uncertainties in dust

simulation. The authors did a good job introducing and comparing the deposition scheme physics in section 2.3 and should be commended for finding/reporting several undocumented discrepancies in the various WRF-Chem model versions. Overall, this paper brings attention to a process that is often overlooked in dust transport model assessment and is of value to the modeling community.

1. However, critical gaps remain in the authors' methodology that need to be addressed before publication moves forward. The current approach supports the authors' assessment that GOCART and Shao2011 produce markedly different dust emission flux patterns, BS95 removed the most dust from the atmosphere, Z01 removed the least dust from the atmosphere, and the S11Z01 combination produced the best simulation of average PM10 and AOD for this case study. These results, however, may only be applicable to this case study and particular WRF-Chem configuration.

For example, the authors did not include or allude to an analysis of how well the model simulated the general meteorological conditions driving the dust events. How well did the model winds (surface and aloft), synoptic conditions, etc. verify against observations? This is a necessary step to be able to discern whether simulation outcomes (good or bad) are actually due to dust scheme physics or an artifact of erroneous forcing conditions.

Many thanks for your comments and constructive suggestions! We agreed it is indeed important to evaluate how well the general meteorological conditions are simulated for a study based on a dust storm event. We now added the evaluation of the meteorological conditions in the revised manuscript. A new figure and a new Table are included (see Fig. 2 and Table 4 in the revised manuscript), and these are discussed in in the beginning of the results part (Sect. 3. 1 P14 L304-324). It shows that the WRF-Chem performed well in simulating the meteorological conditions during our simulation period. So, the differences of simulated dust emission and deposition are attributed to the different emission and dry deposition schemes. Through the evaluation of the meteorological conditions, we have a deeper understanding of the dominant weather system for the dust emission processes, and this help us to identify which dust emission scheme correctly captured the emission phase of this dust event (see our reply below).

2. Also, it is unclear which dust emission scheme correctly captured the emission phase se of this dust event (with respect to magnitude, spatial footprint, and temporal patterns). Assuming the Taklimakan and Gobi Deserts are the primary sources of dust, the scatter plots from Figure 7 seem to indicate GOCART did a better job with dust emission from the Taklimakan Desert and that there's little difference between the results of the two dust emission schemes for the Gobi Desert region. Figure 2 shows maps of the total simulated dust loading from the two dust emission schemes. GOCART clearly produces widespread low-level dust emission, while Shao2011 emits stronger dust plumes from localized sources, mostly from the Gobi Desert region. The AOD comparison in Figure 8 is for midway through the dust event. It is unclear if dust originated in the Taklimakan Desert and was transported, if the Taklimakan Desert region in the AOD observation is cloud obscured, or if the dust in the Gobi Desert is entirely local.

Furthermore, because the authors have chosen to evaluate the dust emission and deposition schemes simultaneously, it's difficult to draw more generalizable conclusions about deposition scheme performance. Z01 may appear to be the best dry deposition scheme, but the slower deposition rate in Z01 may be compensating for the dust emission schemes not producing enough dust in the first place, issues with boundary layer mixing, or simulated winds that were too weak.

Many thanks for the comments and constructive suggestions! We agree it is indeed helpful to better quantify which dust emission schemes may work better for this case, as this will further help to quantify which dry deposition schemes works better. One way for better quantifying how well dust emissions are simulated is to examine how dust emission phase is simulated as the reviewer suggested. We now examined how the daily AOD evolves during the whole dust event period in Fig. 9. The upper panel of Fig. 9 shows the MODIS daily AOD from 1 May to 5 May. We can see relatively small dust emission in the GD on 1 May (Fig. 9a), and the emitted dust is transported eastward on 2 May (Fig. 9b). The main dust emission of this dust storm occurred on 3 May (Fig. 9c), and the dust emission over the GD is really large. Meanwhile, the newly added Fig.S1c and Fig. 2c shows that the Mongolian cyclone on 3 May developed vigorously and caused the large wind speed over the GD. At this stage, the TD has only very small dust emission. Along with the southeast wind and the northeast wind (Fig. S1c, S1e and Fig. 2c, 2e), the dust emission from the GD (Fig.9c) was transported to the northeast and southeast of China. The TD has a very small amount of dust emission, and its contribution to this dust storm is small.

Overall, our results indicate that this dust storm is mainly caused by the huge dust emission from the GD.

Figure 9 shows that Shao2011 well simulated the dust emission process over the GD on 3 May (Fig. 9m), while GOCART obviously underestimated the dust emission over the GD (Fig.9h). Though Fig. 8 (Fig. 7 in the original manuscript) indeed shows that GOCART is better than Shao2011 over the TD region, the dust emission over the TD is very small, and the dust over TD is not easily transported to northeast and Southeast China due to terrain and meteorological conditions.

We also can see that the main dust source regions over the GD are located in the westernmost of Inner Mongolia Province and the southernmost of Mongolia (Fig. 9c). However, from the PM10 observational sites plot (Fig. 7), we can see that in the westernmost of Inner Mongolia Province and the southernmost of Mongolia, there is almost no PM10 observational site. So from the scatter plot (Fig. 8), there is little difference between the results of the two dust emission schemes for the GD. However, away from the dust source region (Fig. 7, NCP,NEP and YR ), Shao2011 is significantly better than GOCART.

Overall, our results indicate that dust emission from Shao2011 is better for this dust event, in terms of dust spatial and temporal distributions. Based on the Shao2011 dust emission scheme, we can then comfortably conclude that Z01 dry deposition scheme performs the best among three dry deposition schemes we evaluated (Fig. 8d,8e,8f). Moreover, even with the GOCART scheme, Figure 8 also showed that Z01 performs the best.

We now incorporate these discussions in the revised manuscript (Sect. 3.4 and Sect. 4).

3. An overview of how well the model (or each simulation depending on whether or not aerosol feedbacks were affecting weather evolution) captured the general atmospheric conditions of the case study event is needed. This could be part of the main text or added as an appendix. This is particularly important given the strong influence of wind flow/turbulence and boundary layer mixing on the deposition process.

Thanks for the suggestion. The evaluation of the meteorological conditions are now added. See our reply above (general comment #1).

4. Dust emission observations are difficult to obtain (or in some cases non-existent). Understanding the evolution of the weather forcing conditions in combination with

the dust emission simulation results, and possibly even qualitative assessment of true or false color satellite imagery, would enable the authors to make inferences as to which dust emission treatment was more accurate for this particular case study. Timeseries plots comparing PM10 observations to simulated PM10 values from grid points in/near the source regions may also offer some insight.

Thanks for the suggestion! We now indeed include the time evolution of simulated AOD and compare them with MODIS observations. This helped us to identify which dust emission scheme is better in simulating this dust event. See our reply above (general comment #2).

5. Given the focus on deposition, this paper really should include an assessment of the simulated vertical dust distribution. For example, the authors could add a comparison of simulated vertical dust distribution to CALIOP LiDAR observations from the CALIPSO satellite (Winkler, 2009; available via the NASA Earth Data Portal at https://search.earthdata.nasa.gov/) in order to demonstrate that simulated dust was in good agreement with observed plume heights before making assumptions about fall rate accuracy.

Thank you for the suggestion. We now added a new figure (Fig.10) to evaluate the vertical dust distribution. The extinction coefficient from CALIPSO is used. We added the description of CALIPSO data (P13 L294-301) and the extinction coefficient evaluation (P21 L477-485). All the six experiments show the similar dust location in the atmosphere, which is consistent with the CALIPSO observation. The simulated extinction coefficients using GOCART dust emission schemes are significantly underestimated compared to the CALIPSO observation (Fig. 10a,10b and 10c), while the modeled vertical extinction coefficients using Shao2011 dust emission scheme agrees better with observation though they are still underestimated (Fig. 10e,10f and 10g). Among all the six experiments, results from S11Z01 agree the best with observation.

Hopefully, these issues can be easily addressed with additional plots and documentation. Papers by Ma et al. (2019), Letcher and LeGrand (2018), Rizza et al. (2017), and Nguyen et al. (2019) offer good examples of approaches for general dust case study descriptions, forcing weather evaluations, and/or vertical dust distribution assessments.

Many thanks for your constructive comments and suggestions. Following your suggestions and these papers, we now added new analysis of the meteorological conditions and the time evolution of the dust storm. We believe these new analyses have greatly improved the manuscript. Please see our detailed reply above.

Specific Comments:

1. P3 L59: "Large-size" is a relative term. Please provide a value for a frame of reference. For example, ". . . large-size aerosol particles (e.g., diameters > X µm), such as dust."

We added the reference value for the large-size aerosol particles **(eg., diameters > 2.5 µm)** (P3 L60).

2. P3 L69, P13 L282, P17 L398, P18 L401: "A lot" is somewhat colloquial for use in an academic paper.

Thanks. We changed our expressions to make it more academic accordingly. Below we list our modifications:
(1) "the dust emission fluxes and surface concentrations **differ a lot**." has been replaced by "**significant difference exist** in the dust emission fluxes and surface concentrations" (P3 L70-71)
(2) "Simulated dust emission fluxes can **differ a lot** between two versions of the Shao2011 scheme" has been replaced by "simulated dust emission fluxes **are quite different** when using two versions of the Shao2011 scheme" (P16 L368).
(3) "as dust emission fluxes in dust source regions **differ a lot** among different dust emission schemes" has been replaced by "as dust emission fluxes in dust source regions **differ substantially** among different dust emission schemes" (P21 L500)
(4) "and simulated dust emission fluxes between WRF-Chem v3.9 and WRF-Chem v3.7.1 can **differ a lot**" has been replaced by "and **significant difference exist in** the simulated dust emission fluxes between WRF-Chem v3.9 and WRF-Chem v3.7.1" (P22 L502-503).

3. P3 L71: Please adjust the text to make it clear that the papers by Yuan et al. (2019) and Chen et al. (2017) are also WRF-Chem studies.

We added "…WRF-Chem simulation of …" to make it clear that the paper by (Yuan et al., 2019) is WRF-Chem study (P4 L75). And we also added "Based on WRF-Chem studies…" to make it clear that the paper by (Chen et al., 2017a) is WRF-Chem study (P4 L77-78).

4. P4 L75: Why is it important to evaluate the dust emission and deposition schemes simultaneously? Wouldn't it be better to select a case study with well-simulated dust emissions from a single dust emission scheme when assessing model sensitivity to deposition scheme configuration? Model performance assessment for different pairings of dust emission and deposition schemes over an extended period of time may be of value to some readers, but evaluating the two aspects of the dust transport process simultaneously for a single case study event introduces extra degrees of freedom that make it difficult to ascribe model performance to a particular root cause.

Thanks for your comments. As we mentioned in our reply to comment #4 from reviewer#1, there are two reasons why we also evaluated dust emission schemes in our manuscript. One is that although there have been several previous studies comparing different dust emission schemes, these studies are all under the framework of the GOCART aerosol scheme, but the dry deposition treatment of GOCART aerosol scheme is problematic for dust simulation based on Zhang et al. (2019). So we do not know yet which dust emission schemes is best for our dust storm simulation based on the previous studies. The other reason is that in the currently released version of WRF Chem, only GOCART dust emission scheme is coupled in MOSAIC aerosol scheme. As shown in our manuscript, the GOCART dust scheme strongly underestimated dust concentrations in comparison with observations, no matter which dry deposition scheme is used. Therefore, we have newly implemented the Shao2011 scheme into MOSAIC aerosol scheme. In the revised manuscript, we also followed the suggestion from the reviewer to examine the evolution of simulated AOD in comparison with MODIS observation for the period of 1-5 May. In combination with the evaluations of meteorological conditions and other aspects of dust simulations, we now can see that Shao2011 indeed performs better for simulating this dust event (see also our reply to general comment #2 from reviewer #2). We believe this comparison therefore indeed provide new knowledge to the field and provides a good reference for the community.

5. P4 L82-84: The authors reference a study by Zhang et al. (2019) that found that the WRF-Chem GOCART model underestimated dry deposition in north-west China by more than an order of magnitude compared to observations. Interestingly, the study by Zhang et al. (2019) was done using WRF-Chem v3.7.1. An error was recently discovered in how the GOCART gravitational settling code was implemented in WRF-Chem that also affects the calculation of the dry deposition rate (see code commit change comment in the WRF source code repository by Alexander Ukhov; https://github.com/openwfm/WRF-Fire-merge/commit/2ffdebf4ac311a5b1ef8cd0c639e0d857b550fdb). Given that this error wasn't corrected until the release of WRF-Chem v4.1, the findings from Zhang et al. (2019) may no longer be representative of GOCART in the current WRF-Chem release. It would be good if the authors note that here for reader awareness.

Thank you very much for the important information. We read the bug fix in GOCART gravitational settling carefully. Before the bug fix, the dust mass is not balanced, with deposited dust and dust in the atmosphere > emitted dust, because there is incorrect dust mass increasing in the gravitational settling subroutine. After the bug fix, the dust mass is balanced, with deposited dust and dust in the atmosphere = emitted dust, which leads to even lower dust deposition after the bug fix compared to that before the bug fix. So after the bug fix, the conclusion from Zhang et al. (2019) that "WRF-Chem GOCART model underestimated dry deposition in north-west China by more than an order of magnitude compared to observations" still holds and the underestimation of dust deposition in WRF-Chem becomes even larger.

6. P5 L101: "The model setups are listed. . ." wording is odd. Suggest changing to "A summary of the settings used to configure the model are listed. . ."

Thanks! This is now updated (P5 L106-107).

7. P5 L103 and Table 1 – Please add the radiation time step to your model description. Simulated wind speeds and dust emission flux appear to be very sensitive to this parameter when using RRTMG (not well documented). Also, please include the land use dataset (lu_index) used for this study in the configuration description given that some of the deposition scheme parameters have dependencies on land use categories.

Thank you for your suggestion. We added the radiation timestep (radt) in the Sect. 2.4 (P12 L254). We added the land category (num_land_cat) in Table1. And we also added the descriptions of land category in Sect. 2.1 (P5 L108-109).

8. P5 L107-109: Suggest combining the following sentences to avoid redundancy: "The MOSAIC aerosol scheme uses sectional approach to represent aerosol size distribution. The MOSAIC 4-bin aerosol scheme divides aerosol particles into four size bins by aerosol diameter: 0.039-0.156, 0.156-0.625, 0.625-2.5, 2.5-10.0 µm." Suggest changing to: "The MOSAIC 4-bin aerosol scheme divides airborne particles into four size bins by their effective diameter (0.039-0.156, 0.156-0.625,0.625-2.5, 2.5-10.0 µm) to represent aerosol size distribution."

Thanks for the suggestion. We combined the two sentences to one accordingly (P5 L114-117).

9. P5 L112: Suggest deleting "from dust emission schemes" to avoid redundancy.

Thanks! This is now removed (P5 L121).

10. P6 L122-123: This statement as written (also stated in other peer-reviewed publications) misrepresents the findings from the Shao et al. (2011) paper. Shao et al. (2011) concluded that their simplified scheme produced similar results to the Shao (2004) scheme when compared to observations from the Japan-Australian Dust Experiment (JADE). The same Shao et al. (2011) paper also notes that these findings shouldn't be generalized due to the conditions of the JADE experiment. Recommend the authors simply note that they chose to use the most simplified version of the University of Cologne (UoC) dust emission schemes for their experiment or confirm Shao2004 and Shao2011 produce similar dust emission flux outcomes for their particular case study.

Thanks. We removed the sentence "but the performances of the full scheme (Shao2004) and the simplified scheme are equally effective (Shao et al., 2011)". As we didn't perform the simulation of Shao2004, we just simply note that we choose to use the most simplified version of the UoC dust emission scheme (P6 L137-140).

11. P6 Section 2.2.1: The GOCART emission scheme description is rather sparse compared to the Shao scheme descriptions. The authors reference the paper on

MOSAIC by Zhao et al. (2010), which offers a similar brief overview of the GOCART dust emission scheme and references the original Ginoux et al. (2001) paper. However, closer examination of the code (subroutine mosaic_source_du in module_mosaic_addemiss.F) indicates dust_opt=13 (at least in v3.9) also includes the modifications to the original GOCART dust emission scheme (dust_opt=1) documented by LeGrand et al. (2019; Section 3.2.1) with the exception of the C parameter (default C value is set to $1 \times 10^{-9}$ kg s2 m−5 consistent with the original Ginoux et al. (2001) paper). It would be good if the authors could expand this section given the general lack of documentation on the WRF-Chem dust_opt=13 setting (currently not included in the WRF-Chem user's manual). Also, the paper by Zhao et al. (2010) explored more than one modal size distribution configuration. It would be beneficial to readers for the authors to describe how the emitted dust particle size distribution used in dust_opt=13 is prescribed.

Thank you for the suggestion. Now we add the description of the dust_opt=13 in detail in Sect. 2.2.1. It reads" the GOCART dust emission scheme within MOSIAC aerosol scheme is called by setting dust_opt=13. " (P6 L129-130)
We added the description of size distribution in Sect 2.2.1. It reads "The original GOCART dust emission scheme in GOCART aerosol scheme (dust_opt=1) calculates the dust emission flux from 0.2 to 20 μm. For GOCART dust scheme in MOSAIC aerosol scheme (dust_opt=13), the total dust emissions from 0.2 to 20 μm are redistributed to the size bins of MOSAIC (0.039-0.156, 0.156-0.625, 0.625-2.5 and 2.5-10.0 μm) with mass fractions of 0%, 0.38%, 8.8%, 68.0%". (P7 L152-155)
And we note here that in addition to the size distribution, the values of empirical proportionality constant C are also different for the two GOCART dust emission scheme options. It reads "We note that in addition to the size distribution, the values of empirical proportionality constant C are also different for the two GOCART dust emission scheme options. For dust_opt=13, C value is set to $1.0 \times 10^{-9}$ kg s$^2$ m$^{-5}$, which is consistent with the original GOCART dust emission scheme paper (Ginoux et al., 2001). For dust_opt=1, C value is set to $0.8 \times 10^{-9}$ kg s$^2$ m$^{-5}$." (P7 L155-159).

12. P6 L142: ". . . we cut the size bins for MOSAIC aerosol scheme from Shao2011 directly." Please be clearer on how the emitted dust size bins were configured for both the GOCART and Shao2011 simulations. The UoC emitted dust size bins have diameter ranges of < 2.5, 2.5–5, 5–10, 10–20 µm in v3.7.1 and 0.2–2, 2–3.6, 3.6–6, 6–12, 12–20 µm in v3.9. UoC emitted dust size bins from v3.9 match the emitted dust size bins from GOCART dust_opt=1. Emitted dust size bins in GOCART

dust_opt=13 appear to be modified to ignore dust particles larger than 10 µm and match the 4-bin distribution used by MOSAIC (MOSAIC bins also noted by authors on P5 L109). Does this statement imply the authors modified the MOSAIC module aerosol size bins to incorporate 5 bins and larger particles (particles up to 20 µm) for the simulations configured with Shao2011? Also, the use of the word "cut" here is a little colloquial.

For GOCART dust emission scheme under MOSAIC framework, it first calculates the total dust emission fluxes (P6 Eq. (1)) and then the total dust emissions from 0.2 to 20 µm are redistributed to the size bins of MOSAIC (0.039-0.156, 0.156-0.625, 0.625-2.5 and 2.5-10.0 µm) with mass fractions of 0%, 0.38%, 8.8%, 68.0%. For shao2011 dust emission scheme under MOSAIC framework, it calculates the emitted dust from 0.98 um to 20 um with 40 size bins. Dust emissions from these 40 size bins are then grouped into the four size bins of the MOSACI aerosol scheme (0.039-0.156, 0.156-0.625, 0.625-2.5, 2.5-10 um). Please also see our reply to specific comment #11 from reviewer #2.

Shao2011 first calculates the emitted dust from 0.98 um to 20 um with 40 size bins. In MOSAIC, dust emissions from these 40 size bins are directly grouped into the four size bins of the MOSAIC aerosol scheme (0.039-0.156, 0.156-0.625, 0.625-2.5, 2.5-10 um). The step that group the size bins to < 2.5, 2.5–5, 5–10, 10–20 µm in v3.7.1 (or 0.2–2, 2–3.6, 3.6–6, 6–12, 12–20 µm in v3.9) is skipped.

We did not modify the MOSAIC module aerosol size bins to incorporate 5 bins and larger particles. For the Shao2011 dust emissions scheme within MOSAIC module, the size bins are also the same as the original MOSIAC size bins (0.039-0.156, 0.156-0.625, 0.625-2.5, 2.5-10 um).

We revised our manuscript accordingly to make the size bins for GOCART (P7 L152-155) and Shao2011 (P7 L169-171) clearer.

13. P7-10 Section 2.3: Please introduce what is meant by important terms like "rebound effect," "collection efficiency from interception," "Schmidt number," and "Stokes number" to help readers understand why differences in these parameters matter for the deposition process.

We added the description for rebound effect in the main text and it reads "When large particles (usually >5 µm) hit the non-sticky surface, they are liable to rebound from the surface if they have sufficient kinetic energy. The rebound factor R represents the fraction of particles that stick to the surface." (P11 L242-244)

We added the description for collection efficiency from interception in the main text and it reads "$E_{IN}$ is the collection efficiency based on the relative dimensions of the particle to the collector diameter (Gallagher, 2002). Interception occurs when particles moving with the mean flow and the distance between an obstacle and particle center is less than half of the diameter. Then the particles will collide with and be collected by the obstacle" (P10 L228-231)

We added the description for collection efficiency from impaction in the main text and it reads "$E_{IM}$ is the collection efficiency due to impaction of the particle with the collecting surface (Gallagher, 2002). Impaction occurs when there are changes in the direction of airflow, and particles that cannot follow the flow will collide with the obstacle and stay on the surface due to the inertia (Giardina and Buffa, 2018)." (P9 L194-196)

We added the description for Stokes number and it reads "St is the ratio of the particle stop distance to the characteristic length of the flow and describes the ability of particles to adopt the fluid velocity." (P9 L200-201)

14. P9 L190 and L196: Please provide ranges for the α and A parameters. Does use of Z01 have a dependency on WRF-Chem being configured with a particular land use dataset given the dependency pf γ, α and A on the land use category (LUC)?

Thank you. We added the ranges for the α (P10 L224) and A parameters (P11 L234) in the main text. Yes, USGS land use categories (num_land_cat=24) should be used when using Z01 dry deposition scheme. As the default setting of land use categories is not USGS since WRF v3.8, we added this important information in Sect. 2.4. It reads "We note here that the USGS LUC should be selected for Z01 dry deposition scheme"(P12 L273-274).

15. P10 Section 2.4: Please include the number of vertical levels used and the time step in the model description section.

Thanks. We added the number of vertical levels and the time step in Sect 2.4. It reads "… and 35 vertical levels with model top pressure at 50hPato simulate the dust storm in May 2017" and " The simulation period is from 26 April to 7 May 2017 with time step of 60s and frequency of output every hour." (P11 L252-253 P12 L254)

16. P10 L214-215: This section needs a figure showing the model domain or a reference to one of the other figures showing the whole model domain. Suggest expanding Figure 4 (WRF-Chem EROD parameter) to include the whole model domain, changing Figure 4 to Figure 1, and referencing the EROD parameter in the dust emission scheme description section.

Thanks for the suggestion. Now we expanded Fig.4 and changed it to Fig.1. And we referenced the erodibility factor (Fig.1) in the GOCART dust emission scheme description sector (P6 L144). We also cited the domain map in Sect. 2.4 (P11 L253).

17. P10 L219-221: The authors note that meteorological conditions are reinitialized every 24 hours, provide two examples of studies that also used this approach, and comment that this approach has been verified to obtain better meteorological fields. The references provided, however, don't really support this statement. For example, Su and Fung (2015) reinitialize their meteorological fields ever 4 days, and neither study explored the use of different "spin-up" approaches on their results. Reinitialization or "daisy-chain" spin-up is a common practice used by numerical weather modelers. As long as the resultant weather fields used in the experiment were representative, the justification statement (L220-221) is unnecessary.

We now evaluated the meteorological conditions (see Fig. 2 and Table 4 and the discussion in the beginning of the result section) (Sect. 3. 1 P14 L303-324). The weather fields are very similar in all experiments and close to the reanalysis dataset we used to drive the model. So we removed the justification statement in the main text.

18. P11 Section 3 (Results): MOSAIC incorporates aerosol feedbacks. The six tests most likely were subject to different weather forcing conditions as the simulations evolved. How notable were those differences?

We evaluated the meteorological conditions in six different experiments and found the difference is very small. Take the weather conditions at 06:00 UTC on 3 May (Figure S2) as an example, we can see that the wind field at 10 meters and temperature at 2 meters are almost the same among six experiments. We also calculated the correlation coefficient and RMSE of wind field at 10 meters and temperature at 2 meters between simulated weather conditions and FNL data (Table

S2). The correlation coefficients for all the experiments are almost the same (Table S2). Although aerosol feedback is turned on, the effect on weather conditions is small, probably because we reinitialized the meteorological condition every 24 hours.
We described the weather forcing difference of the six experiments in Sect. 3.1(P14 L322 P15 L323-324).

19. P11 L241-242: The authors utilize AOD simulation results at 1300 local time to compare with the daily MODIS AOD product. The actual model domain encompasses multiple time zones though (e.g., Fig. 1). Are the model values used for the analysis based on the central point of the model domain (UTC + 8 hours), or was there some other approach used to create a composite simulation product? A comparison of a single simulation time period to the daily product may be fine if the model correctly captured the timing of the forcing conditions. Was this the case? If not, it may be better to compare the daily MODIS product to the simulation time period that best matches the state of the atmosphere when the observations were collected or use simulated daily averaged-AOD values for the comparison.

The model values used for the analysis are not based on the central point of the model domain. We extracted the model results at 13:00 local time for each region. First, we divided the domain into different time zones according to the longitude. Then we can get the UTC when the local time is 13:00 in different time zones. Next, the simulated AOD results are extracted at the corresponding UTC to build an AOD map for the entire domain for each day. Finally, these processed daily model results are compared with the MODIS daily AOD data. For example, for the UTC+8 Time Zone with longitude near 120E, we use the WRF-Chem simulated AOD at 05:00 UTC to compare with the MODIS daily AOD data. And for the UTC+6 Time Zone with longitude near 90E, we use the WRF-Chem simulated AOD at 07:00 UTC to compare with the MODIS daily AOD data.
As the A-Train satellites pass most region of Earth at around 13:30 local time, we think it is more reasonable to use the model results at 13:00 local time in each region to compare with MODIS daily AOD data.
We also revised our description of the comparison method to make it clear (P13 L289-292).

20. P11-12 L245-256: Which dry deposition scheme was used for the dust emission analysis? MOSAIC includes aerosol feedbacks, which could affect the surface winds

driving the dust emission simulation. The authors state that they reinitialized the meteorological conditions every 24 hours, but that could still allow enough time for the forcing conditions to be affected. Were the wind fields the same in both emission scheme tests?

In the revised manuscript, Z01 dry deposition scheme is used for all the dust emission analysis. In our original manuscript, the BS95 dry deposition scheme was used for the Sect. 3.2, the Z01 dry deposition was used for the Appendix B. We now also use the Z01 for dust emission analysis in Sect. 3.2 for consistency in the revised manuscript. When using one dust emission scheme, the dust emission difference between different dry deposition schemes is very small even though dust concentrations are significantly different using different dry deposition schemes. Dry deposition schemes affect dust emission process by influencing forcing conditions through aerosol feedbacks. But this influence is very small in our study, as simulated meteorological fields are very similar among different experiments. Please also see our reply to specific comment #18 from reviewer #2.

21. P12 L258: Why was this time period chosen? Is it the highest magnitude of dust emission for the simulation event?

Yes. This time period is almost the highest magnitude of dust emission for the simulation event. We use this time period as an example to discuss about the reasons for the dust emission flux difference from different dust emission schemes, and we can expect the same results in other time period.

22. P12 L263-264 and L268-269: Unless the authors have altered the code, Shao2011 as implemented in WRF-Chem uses the EROD parameter from the original GOCART dust emission scheme as a mask. Dust emission is permitted where the erodibility factor is greater than zero via a binary (0 or 1) multiplier (e.g., LeGrand et al., 2019; section 3.3; implemented in module_uoc_dust.F). Note, areas classified as zero in the default pre-calculated erodibility factor dataset in WRF-Chem over land are either relatively high points in the terrain (maximum elevation in the surrounding $10\circ \times 10\circ$ area) or determined to have vegetation coverage according to a static 1987 annual average land cover dataset derived from $1\circ \times 1\circ$ resolution AVHRR data (see Kim et al., 2013). Was this erodibility factor masking treatment included in the code implemented by the authors into MOSAIC? If not, this is an important distinction to document.

Yes. The erodibility factor masking treatment is included in the code when we implemented Shao2011 into MOSAIC. We added this information in Sect. 2.2.2. It reads "In Shao2011, the erodibility factor is only used to constrain the potential emission regions. Dust emission is permitted in Shao2011 where the erodibility factor is greater than zero." (P8 L167-168)

23. P12 L257-258 and P36 Figure 3: What grain size(s) were used to diagnose $u*t$ and $ut$ (Fig. 3c through 3f)? Are the dust emission fluxes presented in Fig 3. (g and h) representative of the emission flux for that grain size or the total dust emission flux?

For Shao2011 dust emission scheme, $u_{*t}$ (Fig.5c) is the smallest $u*t$ among all the size bins. For GOCART dust emission scheme, $u_t$ (Fig.5d) is the smallest $u_t$ among all size bins. As $u*t$ and $ut$ decrease with the increasing particle diameter in these two dust emission schemes, $u*t$ and $ut$ are for the largest size bin. Because once the $u_t>u_{*t}\_min$ or $u_{10}> u_t\_min$, there will be dust emissions in this region. The dust emission fluxes presented in Fig.5 is the total dust emission flux (0-10 μm).

24. P12 L269-270: The authors' comment that differences in dust emission flux produced by the two dust emission schemes are due to differences in threshold conditions required for dust emission and differences in formulas and parameters used for calculating dust emission. In other words, the two dust emission schemes are very different from each other and produce different results? This has been well documented in other publications and doesn't add to the discussion. Suggest removing this sentence. The narrative flows into the next paragraph without it.

Yes, the two dust emission schemes have different dust emission mechanisms and are very different from each other. We followed the reviewer's suggestion and removed this sentence (P16 L350-351).

25. P12 L272-275: GOCART is also dependent on mean wind shear. Intermittent turbulence is not considered in the GOCART dust emission process either. Dust emission under low wind speed in GOCART from the Taklimakan Desert is likely due to the threshold velocity error described in LeGrand et al. (2019). The erodibility factor values in the authors' model domain max out at 0.35. The application of the erodibility factor decreases the dust emission flux.

Thanks! We modified our manuscript accordingly. It reads "One reason may be the formula used to calculate the threshold velocity (Eq. (3)). The formula used to calculate threshold velocity is from Marticorena and Bergametti (1995), which was originally designed to calculate threshold friction velocity (see (LeGrand et al., 2019) for details). This inconsistency leads to very small threshold velocity in GOCART, which may result in dust emission at low wind speed." (P16 352-356)

26. P13 L280-285: This is an important aspect of the experimental design and needs to be moved to the methodology section/incorporated into Section 2.2. Details about the Shao2011 configuration used in this study should be consolidated to Section 2.2; they should not split between the main text and the appendix.

Thank you for the suggestions. For the first suggestion, we moved this part into Sect. 2.2.2 accordingly (P8 L174-175). For the second suggestion, if we want to make clear the Shao2011 configuration used in this study, we need to describe the details of Shao2011 dust emission scheme and the difference between different WRF-Chem versions of Shao2011 first. But these descriptions will take up lots of space, readers may get stuck and miss the main point (dry deposition part) of our paper. For those readers who are interested in the Shao2011 scheme and the difference between different versions, they can easily go to Appendix A and Appendix B for more details. In this way, we can ensure the fluency of our paper without losing specific details.

27. P13 L294-295: Please provide a reference for the statement "As desert dust mass is mainly concentrated in the large particle size range. . ." and the upper bound of the range. Is this statement appropriate for all desert regions or just East Asia? Also please include the value of the reference diameter (5 µm) in the main text as well as the figure caption. The "coarse" and "accumulation" characterization of emitted dust from MOSAIC needs to be described prior to this discussion. These are somewhat ambiguous terms in the dust literature. Suggest defining these terms in section 2.2.

Thanks for the suggestion. We added the references for this statement (Kok, 2011; Zhao et al., 2013) (P17 L380). As mentioned in Zhao et al. (2013), the dust size distribution from Kok (2011) (nearly 90% for particle diameter > 2.5 µm) has been evaluated and implemented to North Africa, North America, East Asia and Arabian Peninsula (Chen et al., 2013; Kalenderski et al., 2013; Zhao et al., 2011, 2012).

We included the value of the reference diameter (5 μm) in the main text (P17 L381-382). Now the "coarse" and "accumulation" mode is described in Sect 2.1 (P5 L116-117).

28. P15 L350: Suggest replacing the phrase "better than" with "more physically meaningful" here (also in the abstract). "Better physics" does not always translate to better numerical model simulations.

Thanks! This is now incorporated in the main text (P19 L436-437). In the abstract, we used the phrase "a better physical treatment" which is similar to the phase "more physically meaningful".

29. P16 L373: "Extremely high AOD" is a little too vague here. Are the observed AOD values considered extremely high for this type of event in East Asia? Suggest replacing the sentence intro with "The highest AOD values for this case study were observed in. . ."

Thanks for the suggestion! The text is now updated (P20 L462-463).

30. P17 L376-377: Wording here is a little odd. Suggest changing to "Simulated AOD values from the S11Z01 configuration produced the closest match to the observed daily MODIS AOD with respect to magnitude and spatial pattern (Fig.8g).

Thanks for the suggestion! The text is now updated (P20 L468 P20 L469-470).

31. P17 L382-383: Was the cloud cover an issue? Were there any areas masked out for clouds in the MODIS AOD observations that may have actually been high in dust concentration?

Thanks for the question. Cloud cover can be potentially an issue for MODIS AOD. In the MODIS C6 Deep Blue aerosol retrieval algorithm, the retrieval is not performed for cloud- or snow/ice-contaminated pixels. Because this dust storm is triggered by a strong cyclone system, it is accompanied by cloud systems. In the areas masked out by MODIS, there may be dust under the cloud, or it may be uplifted into the cloud. The comparison between simulated AOD and MODIS AOD just performed for the grid points with valid MODIS AOD retrieval. During the simulation period, large areas

of our domain contain no valid MODIS AOD retrieval (Fig. 9). So this may lead to the low correlation coefficient in Table 6.

32. P18 L408-409: Reference needed for the statement ". . . dust emitted from [the] Gobi Desert is the most important source of dust weather in northern China."

We added the reference Chen et al. (2017b) for this statement (P22 L514).

33. P18 L409-410: The paper by Su and Fung (2015) provides an analysis of a single case study event in East Asia. This study offers valuable information, but a single case study is not sufficient evidence to make general claims about model performance over a region. Recommend removing the statement about the Shao2011 scheme being documented to give better performance than GOCART over East Asia from the text. It's unnecessary.

Thank you for your suggestion. This sentence has been removed (P22 L516-517).

34. P19 L424: Why is there an ellipsis (. . .) in Eq. A2?

Thanks. We deleted the ellipsis (now is Eq.A3) (P23 L535).

35. P19 L431-432, P30 L446-447, L463-464, P33 Table B1, P35 Table B3: Suggest adding a subscript to $\beta$ from Eq. A4 since the $\beta$ symbol is also used to represent a different parameter in one of the deposition schemes.

Thank you for your suggestion. Now we change the $\beta$ symbol to $\beta_0$ in Shao2011 dust emission scheme.

36. P30 Table 3: Columns for dust_opt and dust_schme are unnecessary. The value used to activate the Shao2011 dust emission module by the authors may not be the one used by the WRF-Chem source code managers. Listing an arbitrary setting in Table 3 could cause confusion to readers if this new approach is eventually implemented into the baseline code with different activation options later. Suggest noting that GOCART is dust_opt=13 and that Shao2011 is dust_opt=4 with dust_schme=3 in the text in section 2.2 and removing these columns from the table.

We agree and removed these columns (P35 Table 3). And we added the corresponding options of dust_opt of GOCART (Sect. 2.2 P6 L129-130) and Shao2011(Sect. 2.2 P6 L132-134).

37. P33 Table B1: The values for the third row seem to be missing.

Thanks. We added the number of types of $\eta_{mi}$ in the third row of Table B1.

38. P37 Figure 2: The diameter of the emitted dust is less than 10 µm in the dust_opt=13 version of GOCART. Unless the authors have modified the code, the upper range of the emitted dust size bins from dust_opt=3 is 20 µm (in both v3.7.1 and v3.9).

Please see our reply to specific comment #12 from reviewer #2. For the Shao2011 dust emission scheme within MOSAIC module, the upper range of the emitted dust size bins are also 10 µm.

Noted Typos:

1. P14 L302 and P40 Fig 5: Be consistent with symbol case. The particle diameter is represented by a lower case d in all previous equations.

Thanks. The representation of particle diameter has changed to lower case.

2. P11 L241-242: Use of "p.m." is unnecessary with 24-hour clock time.

Corrected.

3. P12 L253: Use of acronym GD for Gobi Desert before it's been defined (on P13 L277).

Corrected. We now first give the definition of the acronym GD (P15 L330) and then use it.

4. P12 L254: Use of acronym TD for Taklimakan Desert before it's been defined (on P12 L271).

Corrected. We now first give the definition of the acronym TD (P15 L329) and then use it.

5. P16 L372: ". . .with MODIS [is provided] in Fig. 8."

Corrected.

6. Punctuation is an issue. Several commas missing from compound sentences throughout the text.

Corrected.

7. Missing the word "the" before desert names throughout the text.

Improved. We corrected this through the text.

**References**

[revised manuscript text omitted]

---

## Author Response (AR2)

We are grateful to the editor for the valuable comments, which helped us to further improve our manuscript. Below we address the editor's comments, with editor comments are in black and our answers are in purple.

Dear Authors,

Thanks for your revised manuscript, which answers several points raised by the reviewers. However, I do feel that despite the new Table 4, validation of the meteorology, and of wind speed at surface in particular, is still a bit short. Table 4 show skill scores averaged over which domain? R/RMSE are fine, but bias should also be shown. Ideally, in addition to Table 4, a plot showing the bias of simulated wind as compared to a reanalysis product, also highlighting the dust producing regions, would be a clear plus.
We have no observations of dust emissions; wind speed is the closest proxy to "observed" dust emissions, so it is a good idea to spend more time in ensuring that it is indeed correctly simulated.

Kind regards,
Samuel

Dear Editor,

Thanks for your comments.
We agreed it is indeed important to ensure that the surface wind is correctly simulated. Now new panels are added in Fig.2 to show the bias of simulated surface wind as compared to the reanalysis data (Fig. 2c,2f,2i,2l). The dust source regions also highlighted in the new panels. The WRF-Chem model was able to simulate the surface wind speed well over the dust source regions. We added the discussion of the surface wind bias in Sect. 3.1 (P13 L294-299).
It reads "Panels (c)(f)(i)(l) of Fig.2 show the difference of daily mean wind speed at 10 meters between WRF-Chem simulation and NCEP/FNL reanalysis data. The WRF-Chem model was able to simulate the wind speed well over the dust source regions (the Taklimakan Desert and the Gobi Desert) and the eastern and southern China, where the differences were mostly in the range of -2.0 – 2.0 m s$^{-1}$. The wind speed is slightly underestimated near the center of the cyclone (Fig. 2c, 2f, 2i, 2l) and as this is away from dust source regions, we do not expect this underestimation causes large bias in dust emissions."
The skill scores in Table 4 are averaged over whole domain. We added this in the title of Table 4 (P35 L817).

Best,
Yi Zeng

[revised manuscript text omitted]